# Early Detection of Earthquakes Using IoT and Cloud Infrastructure: A Survey

Mohamed S. Abdalzaher [1,*], Moez Krichen [2,3], Derya Yiltas-Kaplan [4], Imed Ben Dhaou [5,6,7] and Wilfried Yves Hamilton Adoni [8,9]

1 Department of Seismology, National Research Institute of Astronomy and Geophysics, Cairo 11421, Egypt
2 Faculty of Computer Science and Information Technology, Al-Baha University, Al-Baha 65528, Saudi Arabia; moez.krichen@redcad.org
3 ReDCAD Laboratory, National School of Engineers of Sfax, University of Sfax, Sfax 3029, Tunisia
4 Department of Computer Engineering, Faculty of Engineering, Istanbul University-Cerrahpaşa, Istanbul 34320, Türkiye
5 Department of Computer Science, Hekma School of Engineering, Computing and Informatics, Dar Al-Hekma University, Jeddah 22246, Saudi Arabia
6 Department of Computing, University of Turku, 20500 Turku, Finland
7 Higher Institute of Computer Sciences and Mathematics, Department of Technology, University of Monastir, Monastir 5000, Tunisia
8 Helmholtz-Zentrum Dresden-Rossendorf, Center for Advanced Systems Understanding, Untermarkt 20, 02826 Görlitz, Germany
9 Helmholtz-Zentrum Dresden-Rossendorf, Helmholtz Institute Freiberg for Resource Technology, Chemnitzer Str. 40, 09599 Freiberg, Germany
* Correspondence: msabdalzaher@nriag.sci.eg

**Abstract:** Earthquake early warning systems (EEWS) are crucial for saving lives in earthquake-prone areas. In this study, we explore the potential of IoT and cloud infrastructure in realizing a sustainable EEWS that is capable of providing early warning to people and coordinating disaster response efforts. To achieve this goal, we provide an overview of the fundamental concepts of seismic waves and associated signal processing. We then present a detailed discussion of the IoT-enabled EEWS, including the use of IoT networks to track the actions taken by various EEWS organizations and the cloud infrastructure to gather data, analyze it, and send alarms when necessary. Furthermore, we present a taxonomy of emerging EEWS approaches using IoT and cloud facilities, which includes the integration of advanced technologies such as machine learning (ML) algorithms, distributed computing, and edge computing. We also elaborate on a generic EEWS architecture that is sustainable and efficient and highlight the importance of considering sustainability in the design of such systems. Additionally, we discuss the role of drones in disaster management and their potential to enhance the effectiveness of EEWS. Furthermore, we provide a summary of the primary verification and validation methods required for the systems under consideration. In addition to the contributions mentioned above, this study also highlights the implications of using IoT and cloud infrastructure in early earthquake detection and disaster management. Our research design involved a comprehensive survey of the existing literature on early earthquake warning systems and the use of IoT and cloud infrastructure. We also conducted a thorough analysis of the taxonomy of emerging EEWS approaches using IoT and cloud facilities and the verification and validation methods required for such systems. Our findings suggest that the use of IoT and cloud infrastructure in early earthquake detection can significantly improve the speed and effectiveness of disaster response efforts, thereby saving lives and reducing the economic impact of earthquakes. Finally, we identify research gaps in this domain and suggest future directions toward achieving a sustainable EEWS. Overall, this study provides valuable insights into the use of IoT and cloud infrastructure in earthquake disaster early detection and emphasizes the importance of sustainability in designing such systems.

**Keywords:** earthquake early warning system (EEWS); disaster; management; internet of things; cloud systems; drones; validation; verification; survey

## 1. Introduction

On 6 February 2023, a Mw 7.8 earthquake destroyed southern and central Turkey as well as northern and western Syria. 37 km (23 miles) west-northwest of Gaziantep is where the epicenter was. The earthquake near Antakya, Hatay Province, peaked with a Mercalli rating of XII (Extreme) as mentioned in [1]. The epicenter of the strong earthquake, followed by the one that occurred on February 6 by nine hours, was 95 km (59 miles) to the north-northeast of the most recent one. There was significant destruction and tens of thousands of deaths. At least 57,300 fatalities had been reported as of 20 March 2023, with more than 50,000 of those occurring in Turkey and more than 7200 in Syria [2].

It was the deadliest natural disaster in the history of Turkey and the largest natural disaster to hit Turkey in modern times since the earthquake in Antioch in 526 [3]. Aside from being the deadliest earthquake since the 1822 Aleppo earthquake in modern-day Syria, it was also the worst earthquake worldwide since the 2010 Haiti earthquake and the sixth deadliest of the twenty-first century [4]. The fourth-costliest earthquakes on record, damages in Turkey were expected to total USD 104 billion, and in Syria, USD 5.1 billion [5].

There are strong correlations between earthquakes, climate changes, and mining activities [6–8]. Many scientists have predicted that the frequency of earthquakes will keep increasing [9]. Figure 1 illustrates the location of earthquakes that occurred in the last 12 months with intensity larger than 7 Moment W-phase (Mww). In [10], the first paper on the idea of earthquake early warning systems (EEWS) in 1985 was presented. These systems are networks of ground-based sensors that alert users when the earth starts to tremble.

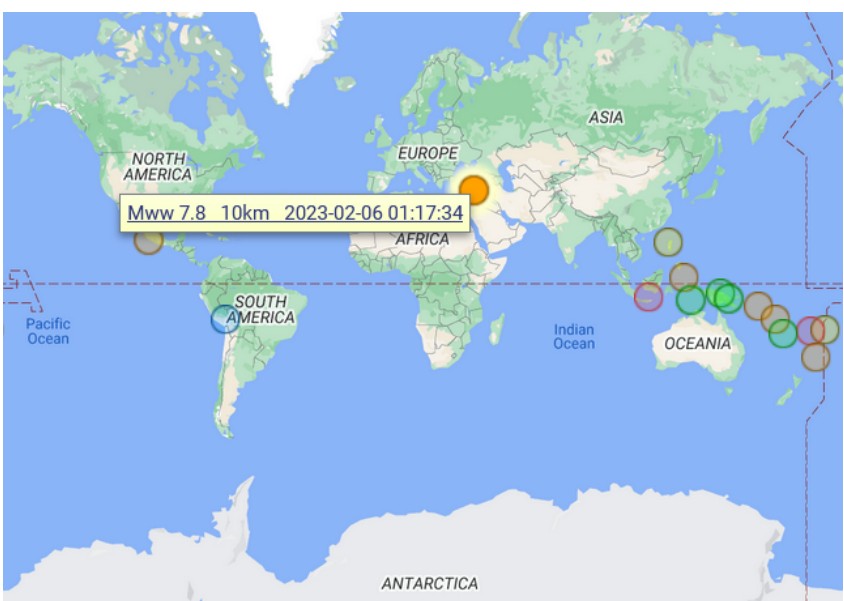

**Figure 1.** Major earthquakes that occurred in the last 12 months as reported in [11].

EEWS operates under the assumption that, despite the slow speed at which seismic waves move, electronic alerts from the epicenter region may be delivered almost instantly. The process is as follows:

1.  Several types of seismic waves radiate from an earthquake's epicenter. Sensors are activated by P-waves, which are weaker but move more quickly. Thereafter, sensors send signals to cloud servers for processing.

2.  Algorithms in the cloud server instantly determine the location, magnitude, and severity of an earthquake. How big is it? Who will suffer from this?
3.  The technology sends out an alert before slower but more destructive S-waves and surface waves arrive.

Those who are close to the epicenter will not receive much, if any, warning beforehand, while those who are farther away could only have a few seconds to brace themselves. EWS may help reduce some of the injuries and damage caused by large earthquakes when used in conjunction with automated countermeasures such as stopping trains or turning off gas lines.

Recent years have seen a significant increase in the number of traditional as well as contemporary technology utilized in EEWSs [12–14]. As a consequence of this, effective integration of the numerous scientific fields is sought after in order to serve such crucial systems. In general, actions taken to reduce risks, conduct seismic hazard assessments, determine site specifications, and the like can be of assistance in this regard [15–19]. Developing a reliable EEWS necessitates solving a number of issues that are impacted by the ongoing difficulties associated with earthquake catastrophes. These issues include the observation of earthquake characteristics, and the environment type [20–25].

Radio-frequency identification, satellite systems, the Internet of Things (IoT), network functions virtualization (NFV), 5G, software-defined networks (SDN), data networks, and a variety of other technologies have all been the focus of significant research in recent years in an effort to lessen the damage that earthquakes cause [26–34]. For instance, satellite systems have been used to track earthquake movements, and IoT sensors have been used to detect earthquakes and provide early warnings. Furthermore, 5G and SDN technologies have been deployed for real-time communication and data transmission in emergency situations. These technologies have greatly enhanced the accuracy and speed of earthquake detection and warning systems and have improved the response time of emergency services.

Moreover, the integration of robots and the internet has the potential to be a significant breakthrough in this field. According to [35], a new integrated system named "robot-event" has been proposed, which is able to execute autonomous inspections and emergency responses to a severe event. The robot uses real-time image tracking to inspect the indoor environment and help any human victims found on the ground. It operates in structurally sound houses with moderate damage, focusing on situations where people are at risk from falling furniture. The system was tested indoors to assess its functionality and operation alongside a smart EEWS. This new technology has the potential to significantly reduce the risk of human casualties during an earthquake by providing timely and accurate information to emergency responders. Future research in this area could explore further the use of robotics, artificial intelligence, and the internet to develop more advanced and efficient EEWSs.

The research conducted in the literature regarding remote sensing applications facilitated by satellite communication systems did not cease with the studies by [36,37]. It also encompassed NFV and SDN, which involved gateways via IoT as well as Micro-Electro-Mechanical systems (MEMS), as noted in [38–41]. The primary objective of this endeavor was to provide relief to areas that had suffered damage or destruction on a large scale. Virtualization played a critical role in this, as it could help mitigate the risks posed by natural disasters. As highlighted in [42], such networks must be designed to optimize node lifetimes. In addition, ref. [43] presented a tragic scenario that showcased an EEWS designed to facilitate a safe evacuation plan against disaster risks by combining cloud-based IoT with heterogeneous networks. Similarly, the combination of IoT and current communication technologies and techniques could prove to be crucial in ensuring the smooth and secure transfer of data, as stated in [44–49].

The studies mentioned here are accompanied by conventional approaches to earthquake detection and analysis, as well as methods for distinguishing between different types of fault ruptures, which have been extensively investigated in the relevant academic literature [50]. In [51], a local similarity earthquake detection approach based on the near-

est neighbor method was proposed to determine whether an earthquake had occurred by examining the consistency of received signals from the nearest neighbors of targeted stations and their closest neighbors. On the other hand, refs. [52,53] focused more on determining the earthquake's amplitude in the first few seconds of its occurrence rather than the complete rupture. However, conventional methods take a significant amount of time to calculate earthquake parameters [54], highlighting the need for additional efforts and studies. Early research suggests that it is possible to accurately predict the magnitude and depth of an earthquake using a graph CNN model that employs batch normalization and attention mechanism techniques. This model can be used in any location with any seismic nodes. The variability of seismic waves and the complexity of the Earth's structure suggest that there is ample room for innovative and adaptable solutions. With the help of modern technologies, the impact of earthquakes on the studied region in [54] can be significantly reduced.

An increasingly useful technology for disaster management is drones. They can be used to gather real-time data and give emergency responders situational awareness, which can aid in improved decision-making and more efficient responses. Drones with cameras and sensors can survey disaster regions swiftly and safely, collecting precise imagery and data that can be used to assess damage, spot areas that require immediate attention, and organize rescue and recovery efforts [55]. Drones can also be used to transport people in hazardous or hard-to-reach locations necessary goods such as food and medical supplies [56]. Drones are an important tool in emergency management because they have the potential to greatly speed up and enhance the efficiency of disaster response efforts.

5G and B5G networks offer several advantages for emergency communication, including faster data transmission speeds, lower latency, and improved reliability [57–59]. These networks can enable real-time communication between emergency responders and affected individuals, as well as the seamless transfer of data and video feeds from IoT devices, such as sensors and drones [60–62].

In particular, D2D communication can play a crucial role in emergency situations, as it allows devices to communicate directly with each other without relying on a centralized network [63,64]. This can be especially useful in scenarios where network infrastructure may be damaged or overloaded, as D2D communication can operate on a peer-to-peer basis and bypass the need for a central network [65,66]. In an earthquake early warning system, D2D communication could allow sensors to share data with each other and trigger alerts in real-time without relying on a centralized system [67,68].

Edge computing can also be leveraged to enhance the performance of earthquake early warning systems [69,70]. By processing data closer to the source, edge computing can reduce the amount of data that needs to be transmitted to centralized servers and enable faster response times [71,72]. For example, in an earthquake early warning system that uses drones to collect data, edge computing could be used to process the data on the drones themselves, rather than transmitting it back to a central server for processing [73,74]. This would not only reduce the amount of data that needs to be transmitted but also enable faster response times in the event of an earthquake [75].

Cloud computing helps manage disasters. Disaster management firms can swiftly deploy vital apps and services to assist emergency response activities using cloud platforms' scalability, flexibility, and accessibility. Cloud-based technologies can monitor and analyze disaster data in real time, helping emergency responders manage resources. Cloud systems can store and handle enormous volumes of data, such as maps, satellite imaging, and social media feeds, to help businesses better analyze disasters. Cloud-based communication and collaboration solutions can also help rescuers communicate, coordinate, and stay linked during the pandemonium. Cloud computing may make disaster management businesses more agile, responsive, and effective in saving lives and minimizing damage. Fog and edge computing are distributed computing methods that provide computing resources near data sources. Fog computing is a dispersed computing infrastructure that processes data near its source. Edge computing brings computing resources to the end-user or device. Fog and

edge computing are important in natural disaster detection and control. Natural disasters damage communication networks, making data collection and transmission difficult. Fog and edge computing provide local data processing and analysis without data centers or clouds. In the aftermath of a disaster, time is of the essence because communication infrastructure may be compromised [76–79].

For example, sensors deployed in an area prone to flooding can collect data on water levels, flow rates, and other factors that can help predict and manage the impact of a flood. By using edge and fog computing, this data can be analyzed in real-time, allowing for EWS to be put in place and emergency responders to be deployed more quickly [80]. Similarly, sensors can be used to detect seismic activity and predict earthquakes, with data processed locally to provide early warning and minimize damage.

Overall, fog and edge computing play an important role in natural disaster detection and management by enabling real-time data processing and analysis at the edge of the network [81]. This approach can help improve the speed and accuracy of disaster response, ultimately leading to better outcomes for affected communities.

Verification and validation (V&V) techniques are crucial for ensuring the quality, reliability, and security of software systems in the context of IoT and cloud computing [82,83]. These systems involve a complex network of devices, sensors, and services that must work together seamlessly and securely [84]. V&V techniques provide a framework for testing and validating these systems, ensuring that they meet the specified requirements and perform as intended. By implementing V&V techniques, developers can identify and correct defects and errors before they cause significant problems, ultimately leading to higher quality and more reliable IoT and cloud systems [85].

In [86], the authors reviewed geospatial and remote sensing technologies in earthquake research and disaster management, analyzing their historical and future applications, limitations, and methodologies. It provides a framework for earthquake hazard, vulnerability, and risk analysis using geospatial technologies. In [87], the study examined remote sensing applications, including Landsat satellite imaging, LiDAR, optical satellite photography, InSAR, and DEMETER in earthquake research. Many other studies [88–90] explore the role of IoT in disaster management and compares IoT-based options for various calamities. It highlights IoT EWS for fires and earthquakes and advises stakeholders on leveraging IoT technology to secure smart cities' infrastructure and minimize risks. The studies also evaluate Caribbean DRM (Disaster and Risk Management) systems, emphasizing the need for technology and new methods in monitoring disaster risks in small island states. It assesses technology in the five DRM pillars and proposes improvements for technology adoption in the Caribbean subregion. The research contributes to the global discussion on technology and innovation in DRM and addresses sustainable development concerns in Caribbean SIDS (Small Islands Developing States).

The review paper [91] explores building damage mapping techniques in post-earthquake scenarios, emphasizing machine learning (ML) and deep learning frameworks. It addresses the drawbacks of manual interpretation of remote sensing imagery and identifies research gaps. The study of [92] reviews remote sensing methods for earthquake risk assessment, highlighting the importance of vulnerability assessment and the need for a comprehensive approach. In [93], satellite remote sensing technology for EEWS is suggested to achieve more improvements for EWS. The research of [94] examines post-earthquake damage investigation using optical remote sensing data and change detection algorithms, discussing their challenges and potential. The authors in [95] analyze how emerging technologies improve disaster management processes and call for further investigation. Lastly, the work done in [96] presents a procedure for managing pre- and post-earthquake stages of structure management using digital tools and emphasizes the role of BIM models and IDM standards.

Compared to previous works, our paper presents a comprehensive overview of the role of EEWS in disaster assessment and relief, specifically focusing on the use of IoT networks and cloud infrastructure. The paper provides fundamental concepts about seismic waves and associated signal processing, details on the EEWS IoT system, and a taxonomy of EEWS

approaches using emerging IoT and cloud facilities. Additionally, our paper elaborates on a generic IoT-enabled EEWS architecture, discusses drones' role in disaster management, and provides a summary of the primary verification and validation (V&V) methods required for the systems under consideration. Finally, the paper describes research gaps in this research domain and provides future directions. While previous papers have discussed geospatial technologies, remote sensing, and social media platforms' use in earthquake research, disaster management, and catastrophe response, our paper focuses on the role of IoT and cloud infrastructure in EEWS and provides a comprehensive overview of the many elements needed to realize an EEWS.

Table 1 is intended to provide a comparison of our work with previous works in the field of earthquake research and disaster management. It summarizes the main focus, methodology, and contributions of each paper, highlighting the unique contributions of our work in relation to other research in this domain.

**Table 1.** Comparison of our work with previous works.

| Ref. | Utilized Technology | Main Focus | Methodology | Contributions |
|------|---------------------|------------|-------------|---------------|
| [86] | Geophysical technology | Earthquake and catastrophe management | Literature review | Earthquake hazard, vulnerability, risk analysis |
| [87] | Remote sensing | Earthquake management | Review of remote sensing applications | Remote sensing pros and cons in earthquake research |
| [88] | UAV hardware | Disaster relief | Field trials and case studies | Implementable framework for drone data collection and analysis for disaster preparedness, response, and recovery |
| [89] | IoT technology | Disaster management | Comparative analysis of IoT-based disaster management options | Practical applications of IoT technology for disaster management |
| [90] | Modern technology | Disaster and risk management | Evaluation of available and applied technology | Suggestions for improving technology adoption across all DRM pillars |
| [91] | Mapping techniques | Mapping in post-earthquake settings | Evaluation of ML and deep learning frameworks | Identification of research gaps and possibilities for real-world scenarios |
| [92] | Remote sensing | Remote sensing data and methods for earthquake risk assessment | Review of remote sensing applications | Necessity for a complete, interdisciplinary approach to earthquake risk assessment |
| [93] | Satellite images | EEWS | Literature review | Evaluation of current and potential applications of remote sensing for seismic disaster early warning |
| [94] | Remote sensing | Post-earthquake damage assessment | Case studies and literature review | Identification of challenges and opportunities in remote sensing for post-earthquake damage assessment |
| [95] | Emerging technologies | Disaster management | Literature review and text mining | Analysis of the effects of emerging technologies on disaster management |
| [96] | Digital tools | Managing existing structures in earthquake settings | Case study | Procedure for managing pre- and post-earthquake stages of existing structure management using digital tools |
| Our Work | IoT nodes and cloud infrastructure | EEWS, environment type, data type, and source, measurement parameters, cloud infrastructure | Literature review and analysis | Comprehensive overview of the role of IoT and cloud infrastructure in EEWS, including a generic architecture and verification and validation methods |

The following is a list of the key contributions that the paper makes, highlighting the various points of innovation:

- We clarify why the EEWS is advantageous for smart cities.
- We emphasize the growth of IoT usage, as well as the IoT system framework in general and its constituent parts.
- We have developed a thorough taxonomy of IoT devices that includes various topics such as the source of data, environment, measured parameters, and factors of validation.
- We present a standard design for the IoT that takes into account potential emergency management.
- We discuss the verification and validation concerns related to using IoT-based EEWS.

The rest of the paper is organized as follows. Section 2 illustrates some generic notions about seismic waves and signals. Section 3 provides an overview of IoT-Cloud systems. Section 4 depicts the IoT and cloud techniques integration in terms of EEWS. Section 5 presents an overview of the verification and validation issues associated with the use of IoT-Cloud-based EEWS. Section 6 lists the main open challenges, concludes the work, and identifies some potential future work directions.

## 2. Seismic Waves and Seismic Signal Processing Techniques

Seismic activity is a key subject of investigation. Understanding how different types of structures respond to earthquake loads and finding out how to safeguard occupants of a structure in an earthquake are both aided by this knowledge.

The study of seismicity can help us better understand the many seismic wave types that are generated, allowing us to map both the regions that are earthquake-prone and those that are not. Studying a region's seismic activity aids in establishing minimum safety requirements for that area, making it simpler for life to go on after an earthquake [97,98].

Acoustic energy, known as a seismic wave, can move through the Earth or another planetary body. It could be caused by a quake (or an earthquake more generally), a volcanic eruption, the movement of magma, a big landslide, or a sizable explosion brought on by human activity, such as mining, which releases low-frequency acoustic energy. Seismologists are responsible for investigating seismic waves. To record the waves, seismologists use accelerometers, hydrophones, or seismometers that are submerged in water [99]. It is important to differentiate seismic waves from seismic noise, also known as ambient vibration, which is characterized by a continuous low-amplitude vibration and can be caused by a wide variety of natural and artificial sources. Arrays of sensors are typically used in seismic signal processing, which is followed by signal conditioning and data fusion. An ADC converter is then used to digitize the gathered data, and a microcontroller is used to process it. This is referred to as an IoT node in the context of the IoT, and it is shown in Figure 2.

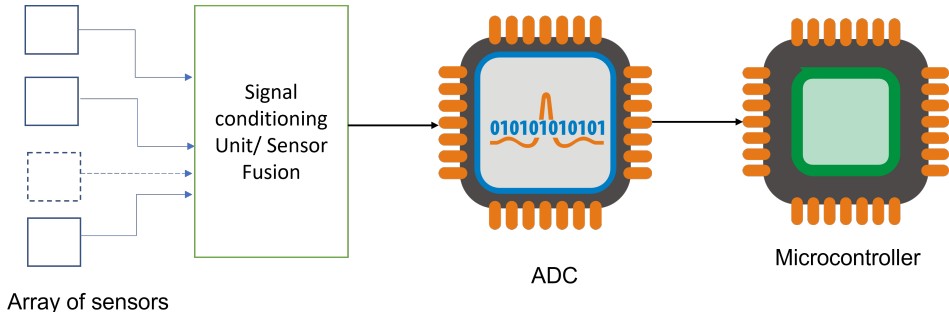

**Figure 2.** IoT sensor node for EEWS.

It is possible to differentiate between the two types of seismic waves known as body waves, which move through the inside of the planet, and surface waves, which move along the surface of the planet. Body waves flow through the interior of the Earth in a manner

that is determined by the paths that are created by material properties such as density and modulus (stiffness). Temperature, chemical composition, and the state of the material all have an effect on the material's modulus and density. This phenomenon can be compared to the refraction of light waves. On the basis of how particles move, body waves can be divided into two distinct categories: primary and secondary waves. Around the year 1830, the French mathematician Siméon Denis Poisson identified this distinction as follows [100]:

- Primary waves, also referred to as P-waves, are longitudinal compressional waves that move through the earth in a straight line. These waves are known as "primary" waves because they arrive first at seismograph stations, traveling faster through the earth than other types of waves. P-waves are pressure waves that can travel through any material, including fluids, and move at a speed that is around 1.7 times faster than that of S-waves. In contrast to S-waves, which are transverse waves that move side-to-side, P-waves are compression waves that cause particles in the material they are traveling through to move back and forth in the direction of the wave's propagation. They take the form of sound waves in the air and move at the same velocity as sound waves, which is around 330 m per second on average. The ability of P-waves to travel through any material allows them to be used to study the interior of the earth. By measuring the time taken for P-waves to travel through the earth from an earthquake's epicenter to a seismograph station, scientists can calculate information about the earth's internal structure. For example, the average speed of P-waves in granite is roughly 5000 m per second, while in water, it is around 1450 m per second. This information can be used to create a detailed model of the Earth's interior.
- S-waves, also known as secondary shear waves, are transverse waves that cause the ground to shift in a direction perpendicular to their propagation during an earthquake. These waves arrive at seismograph stations after P-waves, which are faster. S-waves have a horizontal polarization and move in a horizontal direction, causing the ground to shift from side to side. However, S-waves can only travel through solids since liquids and gases do not support shear forces. They move through any solid medium at a speed that is approximately 60% slower than P-waves. The absence of S-waves in the outer core of the Earth is consistent with the presence of liquid. This is because S-waves cannot propagate through liquids, and their absence indicates that the outer core is predominantly liquid. However, P-waves can propagate through liquids, which is why they can travel through the entire Earth. The study of seismic waves and their behavior has provided scientists with valuable insights into the structure and composition of the Earth's interior.

The path that seismic surface waves take along the surface of the Earth [101]. These are an example of a type of surface wave known as mechanical surface waves. They are referred to as surface waves because their strength decreases as they go away from the ocean's surface. They move at a much slower pace compared to seismic body waves (P and S). The amplitude of surface waves can reach several millimeters during particularly powerful earthquakes.

Seismographs that are situated at a greater distance from the epicenter of an earthquake are unable to detect the high frequencies of the first P wave. In contrast, seismographs that are situated closer to the epicenter are able to record both the P and S waves that are generated when an earthquake takes place [102].

The problems that are associated with seismic data are probably unmatched by any others. During the course of the past few decades, the amounts of such data have nearly multiplied exponentially [103]. In recent acquisition studies, petabits of data are being processed on a daily basis. This requires massive processing capabilities. It should, therefore, not come as a surprise that data formats have evolved significantly over the years and that they have been altered to meet particular workflows or software solutions, which has added to the complexity of managing data [104].

In recent years, the industry of exploration and production has been dealing with "big data" in the form of seismic data [105]. This data is collected during seismic surveys. As

more and more varieties of data are gathered and reprocessed for a variety of purposes, the amount and volume of data continue to grow at an alarming rate. It is necessary to locate and manage both field and prestack data because new insights can be derived from old data by applying updated seismic processing methods. Because of this, it is important to keep track of both sets of data. Several companies made the decision to store this information on tapes because of the massive size of seismic data files and the prohibitively expensive cost of disk space. However, tapes were difficult to handle and regularly went missing, so this was not an ideal solution. Web-based viewers and administration tools make it easier to discover and handle data from anywhere in the world. At the same time, tiered storage and cloud storage offer new and more cost-effective means of keeping enormous seismic datasets [106]. Figure 3 shows the enhancements of the utilized earthquake measurement.

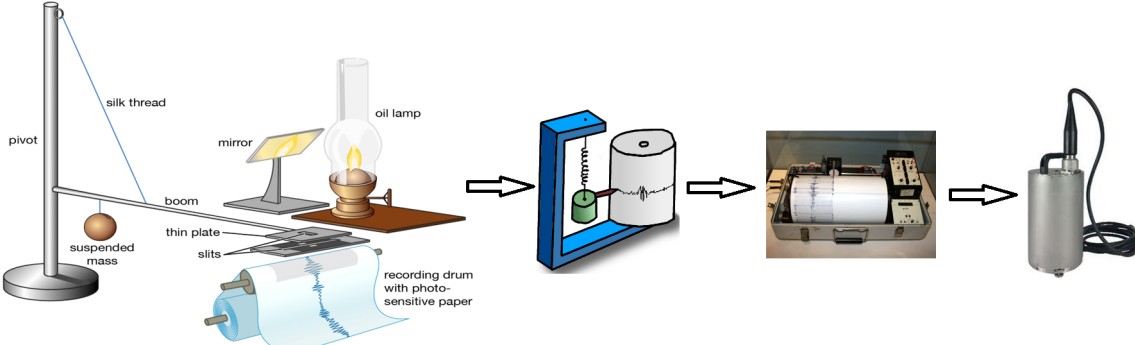

**Figure 3.** Earthquake measurement evolution.

Seismic wave analysis is a key component of earthquake early warning systems, as it enables the detection and characterization of seismic waves in real-time [107,108]. One of the most widely used signal processing techniques in seismic wave analysis is the Fourier transform, which is used to transform time-domain signals into frequency-domain signals [109,110]. In earthquake early warning systems, the Fourier transform is often used to analyze the spectral content of seismic waves, which can provide important information about the location, magnitude, and duration of an earthquake [111]. The Fourier transform is also used to filter out noise and unwanted signals from seismic data, improving the accuracy of earthquake detection and analysis [112].

Another advanced signal processing technique used in seismic wave analysis is wavelet analysis, which is used to analyze signals that are both time-varying and non-stationary [113–115]. In earthquake early warning systems, wavelet analysis is often used to detect and analyze seismic waves that have complex frequency components, such as those generated by slow earthquakes or volcanic activity [116,117]. By decomposing a seismic waveform into its constituent frequency components, wavelet analysis can provide more detailed information on the characteristics of seismic waves, such as their frequency content, duration, and amplitude [118,119].

In addition to these advanced signal processing techniques, earthquake early warning systems also rely on a variety of specific parameters to optimize their performance [120,121]. These parameters include sampling rates, window sizes, and filter cutoff frequencies, among others. Sampling rates determine how often seismic data is collected and stored, while window sizes determine the length of time over which seismic data is analyzed. Filter cutoff frequencies determine which frequency components of a seismic waveform are analyzed and are often used to remove noise and unwanted signals from seismic data.

In conclusion, earthquake early warning systems rely on a variety of advanced signal processing techniques and specific parameters to detect and analyze seismic waves in real time. By providing more detailed information on these techniques and parameters, we aim to enhance the technical rigor of our paper and improve the understanding of the underlying technology. By optimizing the performance of earthquake early warning

systems through advanced signal processing techniques and specific parameters, we can improve the accuracy and effectiveness of these systems, ultimately helping to save lives and reduce the impact of earthquakes on communities.

## 3. IoT-Cloud Systems

### 3.1. IoT Systems

The IoT is gaining increasing support as a viable new technology throughout the world [122,123]. An IoT is a system that relies on connected embedded items or gadgets that have identifiers and are able to interact with one another without the assistance of humans using a common communication protocol. It has been reported that there are more internet-connected devices on the earth than there are humans, where these devices support the smart cities establishment [124–126]. Some smart cities are already in existence [127]. The growth of intelligent technology is outlined on Statista's website [128] under Figure 4. As has been suggested, an enormous increase in smart homes as well as commercial buildings, and important requirements for these buildings will include intelligent electricity and water management [129]. Statista projects that the number of IoT devices will nearly triple between 2020 and 2030, going from 9.7 billion in 2020 to more than 29 billion in 2030. Over 5 billion consumer IoT devices are expected to exist in China by 2030. Accordingly, it is the nation with the majority of these devices. Consumer markets make use of IoT devices; nevertheless, it is anticipated that the consumer market will account for more than 60 percent of all IoT-connected devices by the year 2020 [128]. For the next decade, it is anticipated that this proportion will not change from its current value.

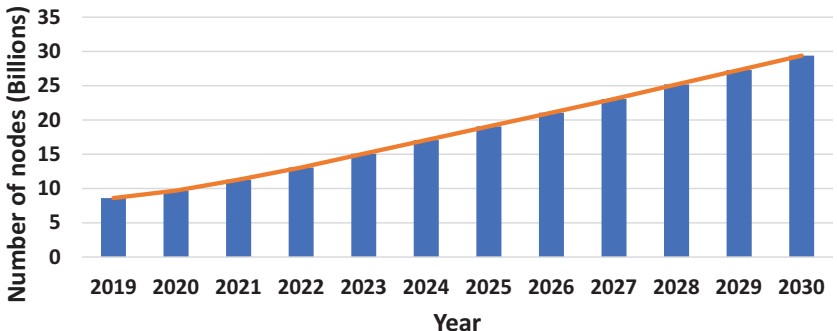

**Figure 4.** Estimated growth of IoT nodes.

According to [130–134], several industry verticals now have more than 100 million connected IoT devices, including government, retail and wholesale, transportation and storage, electricity, steam, gas, air conditioning, waste management, water supply, and retail and wholesale. It is predicted that by 2030, over 8,000,000 IoT nodes will be employed in all industries [128]. In addition, cell phones can represent the best contributor to IoT nodes. Interestingly, it is expected to reach nearly 17 billion by 2030, and more than one billion would be used to connected (autonomous) vehicles [128], information technology infrastructure, asset tracking and monitoring, and smart grids [135–137].

While configuring an IoT system, the following steps should be carried out in accordance with established industry standards [138–143]:

- Providing the node with an interface that can collect data from the environment.
- Providing a tool for acquiring and analyzing data in order to derive knowledge from it.
- Taking action and communicating choices and information to the appropriate hubs.

To gain a comprehensive understanding of an IoT solution's architecture, it is necessary to examine multiple IoT systems. As shown in Figure 5, an IoT system's framework typically consists of a sensor network that monitors changes in the surrounding environment. Depending on the required transmission speed and distance, the collected stream should be transmitted to a centralized or decentralized administration using, e.g., Zigbee, Bluetooth, Tmote Sky, 4G, etc. It is important to note that the sensor system needs

a continuous source of electricity, and the choice of connectivity is influenced by power consumption, with mobile service requiring more power than WiFi, 474.67 to 576.64 mW and 1254.3 to 1540.6 mW, respectively [144]. Safety considerations for both hardware and connectivity are also crucial. An IoT system's data is reviewed or saved in the cloud system to identify patterns and extrapolate information, a critical requirement for any IoT system. The data can be simplified using data visualization, and alert systems can be implemented to provide appropriate levels of caution to users. It is essential that IoT system design is not limited to industry professionals only.

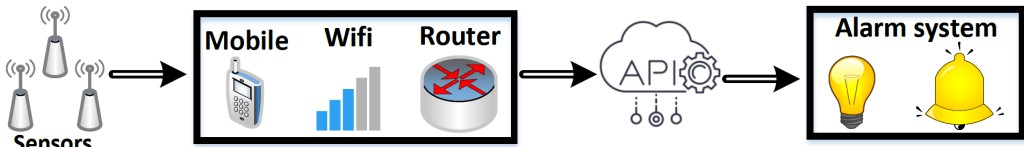

**Figure 5.** A general IoT system paradigm.

The IoT has the potential to revolutionize methods of detecting and managing disasters. With the help of IoT devices, we can collect real-time data on various environmental factors such as temperature, humidity, air pressure, and wind speed, which can help us detect natural disasters such as hurricanes, floods, and earthquakes. These devices can also monitor infrastructure such as bridges, dams, and buildings for any signs of damage or weakness and alert authorities before they collapse or fail, preventing further damage and loss of life.

Moreover, IoT can aid disaster management by providing real-time updates on the affected areas, helping authorities plan and allocate resources effectively. Smart sensors and cameras can be deployed to assess the extent of the damage in disaster-stricken areas, and drones can be used to reach inaccessible areas and gather more information. This data can be analyzed using ML algorithms to identify patterns and predict future disasters [145,146], improving the accuracy of EWS and minimizing the impact of disasters.

Another significant advantage of IoT in disaster management is its ability to facilitate communication between emergency responders and victims. Wearable devices and mobile apps can help victims send alerts and SOS messages, and responders can use IoT devices to locate and rescue survivors in real time. IoT can also help in tracking the movements of rescue teams and ensuring their safety.

The use of IoT for disaster detection and management has the potential to save countless lives and minimize the impact of disasters. By leveraging IoT devices to collect real-time data, authorities can detect disasters early, manage resources effectively, and respond quickly to save lives. However, it is crucial to address concerns about data privacy and security to ensure the safe and ethical use of IoT in disaster management.

Drones, also known as unmanned aerial vehicles (UAVs), have come a long way since their inception in the early 20th century [147]. Initially used for military purposes, drones have evolved and diversified over time, and today they have a wide range of applications in various fields, including photography, agriculture, search and rescue, and disaster management [148]. In addition, it can be equipped with thermal imaging sensors that can also be used to detect the presence of survivors in collapsed buildings or other hard-to-reach areas [149].

There are several types of drones (Figure 6), each with unique characteristics and capabilities. The most common types of drones are fixed-wing, rotary-wing, and hybrid drones. Fixed-wing drones are similar to airplanes and can fly for longer distances, while rotary-wing drones, also known as quadcopters, are more agile and can hover in place. Hybrid drones combine features of both fixed-wing and rotary-wing drones, providing a balance between endurance and agility.

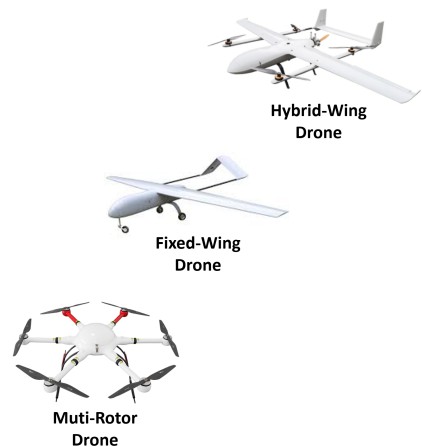

**Figure 6.** Different types of UAVs.

Drones can be involved in various types of communications (Figure 7), including visual, auditory, and data communication. Visual communication involves transmitting images and videos captured by the drone's camera to a remote operator or a ground station. Auditory communication can include transmitting audio messages, such as warnings or instructions, to individuals or groups on the ground via a speaker on the drone. Data communication involves transmitting data, such as telemetry and sensor readings, between the drone and a ground station or another device. Additionally, drones can be equipped with communication technologies such as satellite communication, Wi-Fi, and cellular networks to enable long-range communication and control. In [150], the authors exploited the UAVs to collect data for disaster management relying on 5G (fifth generation) and B5G (beyond 5G) systems with their huge capacity in terms of different data types. The authors investigated various literature solutions to some UAV issues, such as energy harvesting and security. For example, they mentioned a sample method that used UAVs to find the most suitable localization of the sensor nodes for optimizing Quality of Service.

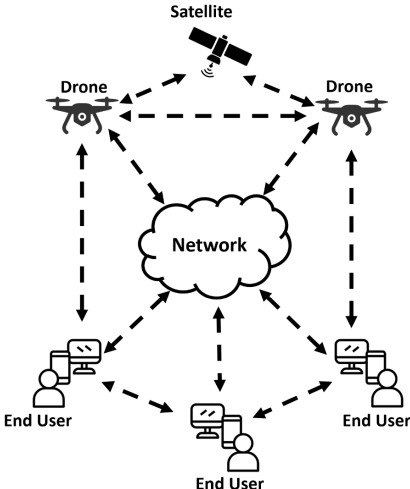

**Figure 7.** Possible types of communications between UAVs and end-users.

In recent years, advances in technology have enabled the integration of new sensing and data collection methods into EEWS systems, including the use of drones [151,152]. Drone-based sensing can provide high-resolution data on earthquake characteristics, such as ground motion and deformation, which can improve the accuracy and effectiveness of EEWS systems [153,154]. The main advantages and limitations of the use of drones is summarized in Table 2.

**Table 2.** Main Advantages and Limitations of Drones.

| Advantages | Limitations |
| --- | --- |
| Good performance in autonomous processes | Requirement of continuous connectivity with the controllers, network coordination |
| Long-distance flights, despite the need for line-of-sight, thus large coverage area | Range limitation proportional to the physical capabilities such as radio controller's range, line-of-sight, and positioning |
| Transmission of big data to the cloud | Limited ability for intelligent data processing |
| Fast-deployed, flexible, and on-demand operative structure | Modeling complexity |
| Low-cost values | The necessity of Quality of Service optimization |
| Usage in dangerous areas | Security challenges such as hijacking |

Real-time communication is also essential for the timely dissemination of earthquake alerts, and various communication technologies, including satellite and wireless networks, are used to transmit sensor data and alerts to processing centers and end-users [155]. In addition, the integration of EEWS systems with IoT and cloud infrastructure can provide scalability, fault-tolerance, and data processing capabilities [156]. IoT devices, such as accelerometers and GPS sensors, can provide additional data sources for earthquake monitoring and analysis. In contrast, cloud infrastructure can provide storage and processing capabilities for large-scale data analysis and modeling [157]. Coordination between these various technical aspects is essential for the successful deployment and operation of EEWS systems, and careful consideration of their capabilities, limitations, and interdependencies is necessary to ensure their effectiveness in mitigating the impact of earthquakes [158].

### 3.2. Cloud and Fog Systems

Cloud computing is a model of delivering computing resources, such as servers, storage, databases, and software, over the internet on an on-demand basis [159]. It provides users with easy access to a wide range of computing resources that can be scaled up or down based on demand without requiring users to invest in and maintain their own physical infrastructure. Fog computing, on the other hand, is a distributed computing model that brings computing resources closer to the edge of the network, closer to where data is generated and consumed, and provides real-time processing and decision-making capabilities [160].

The importance of cloud and fog computing in natural disaster detection and management cannot be overstated. Natural disasters such as hurricanes, earthquakes, floods, and wildfires can cause widespread devastation and loss of life. The use of cloud and fog computing in disaster management can help to mitigate the effects of these disasters by providing real-time data analysis, decision-making, and communication capabilities [161].

Cloud computing can be used to store and process large amounts of data generated by sensors and other devices used in disaster management. This data can be analyzed in real-time, providing EWS to alert authorities and the public of impending disasters. Cloud computing can also be used to store and share critical data such as emergency response plans, evacuation routes, and contact information for emergency services.

Fog computing can be used to process and analyze data at the edge of the network, near the source of the data. This can provide real-time information about the status of infrastructure such as roads, bridges, and buildings, allowing authorities to make informed decisions about evacuation and emergency response efforts. Fog computing can also be used to provide real-time communication capabilities, allowing emergency services to coordinate their efforts and communicate with each other and the public in real time.

In summary, cloud and fog computing play a critical role in natural disaster detection and management. They provide real-time data analysis, decision-making, and commu-

nication capabilities, allowing authorities to respond quickly and effectively to disasters, potentially saving countless lives and minimizing the damage caused by these events.

## 4. IoT-Cloud-Based EEWS

This section will highlight the significant significance that the IoT-Cloud technology plays in EEWS. In point of fact, the application of IoT-Cloud strategies has been of assistance to EEWS before and after disasters.

The IoT has revolutionized the way we interact with the physical world, and one of its most promising applications is in the detection and prediction of natural disasters such as earthquakes. The basic idea behind using IoT for earthquake detection is to deploy a network of sensors that can detect seismic activity and transmit the data to a central server for analysis. These sensors can be embedded in buildings, bridges, and other structures, as well as in the ground itself. By analyzing the data from these sensors, it is possible to detect the onset of an earthquake and predict its magnitude and location.

One of the key advantages of using IoT for earthquake detection is that it allows for real-time monitoring of seismic activity. Traditional methods of earthquake detection rely on seismometers, which are expensive and require a lot of maintenance. They also typically only provide data after an earthquake has already occurred. In contrast, IoT sensors can provide continuous data in real-time, allowing for EWS to be put in place [162–164]. This can be particularly useful in areas prone to earthquakes, where early warning can save lives and reduce damage.

Another advantage of using IoT for earthquake detection is that it can provide more denser data network than traditional methods. IoT sensors can be placed in a wider variety of locations, such as inside buildings or underground, allowing for a more comprehensive picture of seismic activity. They can also provide data on other factors that can affect the impact of an earthquake, such as soil conditions and building materials. This information can be used to develop better models for earthquake prediction and to design buildings and infrastructure that are more resistant to seismic activity.

The use of IoT for earthquake detection has the potential to revolutionize the way we prepare for and respond to earthquakes. By providing real-time data and more comprehensive information on seismic activity, IoT sensors can improve our ability to predict earthquakes and minimize their impact. As the technology continues to develop, we can expect to see more widespread deployment of IoT sensors and more sophisticated analysis techniques, leading to even better earthquake detection and prediction capabilities.

A generic EEWS architecture typically consists of three main components: the seismic network, the processing center, and the alert distribution system [165]. The seismic network comprises a set of sensors deployed across a region of interest, which detect and record seismic waves generated by earthquakes. The sensor data is transmitted to the processing center, where it is analyzed in real-time using algorithms and models to estimate the location, magnitude, and other characteristics of the earthquake [166]. The alert distribution system then disseminates the earthquake alert to end-users through various channels, such as mobile devices, sirens, and public announcements [167]. The underlying infrastructure of the EEWS includes a variety of hardware and software components, including seismometers, communication networks, computing systems, and databases [168]. The seismometers are typically deployed in a dense network to ensure high spatial resolution and coverage, and they are connected to a communication network that transmits the sensor data to the processing center [169]. The processing center comprises a set of computing systems that perform real-time data analysis, using a variety of algorithms and models to estimate the earthquake parameters [170]. The alert distribution system includes a set of communication channels and protocols that disseminate the alert to end-users, as well as a database that stores historical and real-time earthquake data [171]. The interactions between these components are tightly coordinated to ensure timely and accurate earthquake alerts, which can help to mitigate the impact of earthquakes and save lives [172].

In [173], the authors developed CrowdQuake, a DL-based seismic detection system. Utilizing a dense IoT network composed of MEMS nodes, the system employs a multi-head convolution neural network to analyze a large quantity of observed acceleration data. During the model validation procedure, the scientists got data from the National Research Institute for Earth Science and Disaster Prevention (NIED) and measured the precision-recall, accuracy, and noise level. The developed system could process data from up to 8000 IoT sensors, and identifying an earthquake required only a few seconds of processing time, according to the researchers. In [174], an advanced EEWS supported by an IoT network that operates on the basis of real-time alerts has been established. The network utilized MEMS accelerometers and an Arduino Cortex M4 CPU for measuring acceleration. This technique employs ML to improve the accuracy and latency in earthquake detection. The model was constructed using data gathered locally by the MEMS accelerometer nodes that were installed.

In [175], IoT acceleration nodes were designed explicitly for earthquake detection. Two methods are used to utilize these nodes: a technique of standalone and a technique of client-server. The first technique is more commonly used, while the client-server technique is more precise but requires high-performance servers and network infrastructure to manage data acceleration from multiple client machines. Basic earthquake detection methods can be independently explored on less capable mobile nodes. However, this may result in false alarms. To overcome this limitation, a cooperative method that uses a large number of mobile phones located in close proximity to one another is employed. This creates a seismic network that can detect earthquakes and monitors any shaking caused by human activity, mechanical vibrations, earthquakes, etc. By relying on a primary neural network, a motion similar to an earthquake detected by a smartphone is transmitted to other cellphones in the immediate area using a multi-hop mode. Furthermore, every mobile phone in the network determines and notifies the network of an earthquake, then triggers an alarm after obtaining detection data from other smartphones in its immediate vicinity. This technique improves the earthquake detection capabilities of a standalone method that does not use any system or network infrastructures.

In [176], a predictive model that combines IoT devices and ML techniques was used to detect geological landslide occurrences. The predictive model was trained with geotechnical parameters such as soil shear strength, soil moisture, rain intensity, terrain slope, and more. The actual hardware used for this purpose consisted of a collection of sensors that gathered real-time information on the topography and soil. In [177], the authors proposed a compute offloading system architecture that can be implemented on Internet-connected drones. They conducted an in-depth experimental study to compare the efficiency of cloud computing offloading strategy with that of the edge computing strategy for DL solutions in the context of unmanned aerial vehicles (UAVs). The authors investigated the balance between the computational cost of the two alternative options communications in an experiment.

In [53], a DL paradigm based on integrating autoencoder (AE) and CNN was developed to immediately determine earthquake magnitude and position three seconds after the P-wave begins. The authors referred to it as CNN and 3s AE (3S-AE-CNN). The data set used in the study was monitored by three stations of the Hi-net seismic network in Japan, and the approach was evaluated using data from 12,200 separate occurrences (109.80 thousand 3 s three-component seismic windows). The model simplifies the extraction of essential waveform properties, resulting in a higher degree of credibility in earthquake parameter assessment. The suggested model predicts magnitude, latitude, and longitude with an accuracy of within $28 \times 10^{-6}$, $3.3 \times 10^{-6}$, and $100 \times 10^{-6}$ degrees, respectively. That model immediately communicates event features to a sink IoT node. It provides guidance to the relevant administration on how to proceed. It is noted that AE has proved beneficial in feature extraction regardless of the application [178].

The framework for earthquake prediction proposed in [179] is a novel approach based on federated learning (FL). This FL framework outperformed the previously developed ML model for earthquake estimation through an IoT gateway in terms of reliability and

accuracy. The model achieved an accuracy of 88% by analyzing multidimensional data over a 100 km radial area, excluding the Western Himalayas, and studying the data. In [180], an EEW based on an IoT and an ML model was suggested to predict tsunamis using tsunami data dating back to 2100 BC and was trained on earthquake parameters in the dataset. It achieved an accuracy of 95% in predicting earthquake location, depth, and magnitude.

In [181], a DL approach that can identify P-waves despite background noise was developed using MEMS for observing events. The model can detect the probability of occurring preceding significant shocks and accurately predict P-waves between 1.5 and 2.5 s before their arrival. In [182], the authors used detector nodes to detect earthquakes locally by probing the environment and assessing data from probes in the surrounding area. The method stores all data locally, making it resistant to node failures and partial network outages, thus increasing privacy. The test network consisted of twenty node codes joined with ten neighbor nodes chosen at random. The total number of detectors was sampled every ten seconds. In [183], a Multilayer Perceptron-classifier was developed to provide a severity-based warning by predicting the possibility of an onsite intensity exceeding a pre-trained PGA threshold associated with damaging intensities on the MMI scale—seismic properties observed by the strong-motion signal starting from the P-wave in the developed model. The authors of [184] proposed an independent model of earthquake detection via low-cost acceleration nodes. The model utilized four different sensor types for establishing an EEWS with different types of data, e.g., noise from buildings, vibrations, and earthquake records. To test the sensors, two actual earthquakes were replicated on a shake table. The study found that low-cost acceleration sensors can detect earthquakes by monitoring differences in acceleration induced by a range between 0.02 g to 0.8 g, which can be detected by the sensors. Therefore, the authors used scaled data within that range.

An ML methodology with earthquake characteristics was utilized, as opposed to the more conventional seismic methodologies that are typically used [185–187]. The authors broke the detection problem down into two distinct groups, namely, static settings and dynamic settings. They provide the most effective ML approach and input data for the static environment based on an experimental evaluation of numerous features for circumventing the issue of discriminating earthquake and noise components to reduce the number of false alarms. This model was validated with the help of 385 earthquakes ranging in magnitude from 4.0 to 8.0.

The authors of the paper [188] introduced the Distributed Multi-Sensor Earthquake Early Warning (DMSEEW) system as a cutting-edge ML-based technique that includes data from GPS stations and earthquake sensors in order to recognize large and medium earthquakes. The model relies on an innovative stacking ensemble technique that has been validated by geoscientists using a real-world dataset. This approach was used to build DMSEEW. The architecture of the system was designed to be regionally spread, which allows for both brisk processing and resistance to disruptions in some aspects of the underlying infrastructure. To be more specific, these systems combined GPS and seismic data in order to enhance earthquake detection, which led to the creation of an efficient EEWS.

Karacı [189] used vibration sensors for earthquake detection according to some threshold values. If an earthquake has been detected via the vibration sensors, a warning system takes place. In this warning system, there is a Wi-Fi module for Internet connection to send a tweet by means of the ThingSpeak IoT analytics platform service. There is also a sound alarm via a buzzer for the people staying around the earthquake area. Thus, the electronic part of this study covers an Arduino card, Wi-Fi module, Inertial Measurement Unit sensor, vibration sensor card, and buzzer. The software part involves the codes of Processing with Arduino to obtain the sensor data normalization and the difference between sequential sensor values for monitoring the threshold level.

Babu and Rajan [190] have studied an IoT solution that alerts for a flood or earthquake detection before they happen and living beings being searched for during the disasters. Sensors are connected to a microcontroller, RF transmitter, and receiver. Their values are

analyzed in real-time via ThingSpeak IoT analytics platform service [190]. The authorities are notified by GSM messages during a flood or earthquake disaster using the IP protocol. A water level sensor has been used for flood detection, and four color bulbs represent the danger levels. A rain sensor has worked to determine whether there is rain. Additionally, a vibration sensor has been used for an earthquake. The system and the mobile phones are charged with solar energy as a secure option for flood environments. ESP Wi-Fi module is the gateway as the fundamental processing and storage part between the RF receiver-transmitter-bulb system and ThingSpeak cloud server for sensor data transmission. This data can be followed using a ThingSpeak API that is used for mobile phones, laptops, or any other internet-connected device. Wireless communication, especially GPS, is used for living being searches.

Won et al. [191] proposed a high-fidelity vibration sensor consisting of a MEMS accelerometer with high sampling frequency and digital filtering. During the sensing process, Short-Term Average/Long-Term Average trigger is compared with a threshold value. Over the threshold, data acquisition, low-pass filtering, and downsampling to a frequency are performed. After this procedure, if an earthquake is detected with the proposed algorithm, the system notifies it through a Bluetooth Beacon. The authors mentioned that the hardware platform was Adafruit nRF52840 Feather Express developed based on nRF52840 (Nordic, 2019) board having fast computing and high storage attributes provided by CPU, RAM, and flash parts. Additionally, the nRF52840 module has Bluetooth 5 and Arduino IDE support.

Duggal et al. [192] mentioned that the literature studies could not separate any other vibrational noise properly from that of earthquakes. They proposed a new method by using IoT to eliminate this drawback. A Micro Electro-mechanical system sensor is set inside a building after finding the most suitable place via structural analysis with the information on seismic shear walls. Inside this sensor, there is an accelerometer and a gyroscope. The gyroscope saves the ground's shaking pattern, representing a distinctive nature during an earthquake. IoT I$^2$C Communication Protocol is used between the devices in the network. Arduino Uno microcontroller board and NodeMCU Dev Kit firmware have interfaces with the sensor node. The sensor data is sent by Arduino inbuilt WiFi to the ML side. In this part, Logistic Regression, Support Vector Machine, and Convolutional Neural Networks have been used for modeling.

Sharma et al. [89] gave a table for ten different IoT disaster management systems based on several properties, such as IoT architecture ownership, cloud-enabled, computer technology area, main focus, and disaster type. They also classified IoT-based disaster recovery systems into four groups: Service-oriented, natural, artificial, and post-disaster. They compared IoT-enabled disaster management methods according to their wireless communication technologies, sensor types, and some additional features. They also put a comparison diagram exhibiting that Bluetooth and Wi-Fi were the best cost, and Bluetooth and ZigBee had the best power usage among the IoT communication technologies. The authors proposed case studies for forest fire detection and EEWS based on IoT devices. For the earthquake warning part, they mentioned the usage of Vibration Sensors (Accelerometer), PIC (Peripheral Device Controllers), ZigBee communication procedures, LCD monitors, and RS232 cables. An IoT alarm message was sent to smartphones, and an alarm message via GSM standard was sent to other cell phones.

Mishra et al. [193] have optimized a schedule for distributing relief items using IoT technologies, such as smart cities. There are some dynamic features dependent on the disaster conditions, such as changing relief demand and resource availability. IoT is suitable for such issues related to continuously flowing and dynamically changing data [194]. The authors symbolized different time periods with sliding time windows in which the data update occurs. For the first window, relief distribution is decided according to the availability of vehicles, relief resources, priority of the disaster area, and delivery routes. The distribution schedule has been optimized repeatedly in the next time slots.

The fragility of the problem that is being targeted, as well as its direct effect on human life, makes it imperative that a solution be found that is intelligent, trustworthy, and flexible despite the considerable efforts that have been put into developing the state-of-the-art. In this section, we throw light on the primary research explorations that have been done in this area. The primary efforts in developing IoT for the EEWS are outlined in detail in Table 3.

**Table 3.** IoT-based EEWS main efforts.

| Ref. | Sensor Node | Employed Environment | Used Data Type | Used Measurement Parameter | Source |
|---|---|---|---|---|---|
| [184] | Acceleration sensors (MMA8452, LIS3DHH, ADXL355, and MPU9250) | UG | Acceleration data | PGA | NIED and USGS |
| [180] | Mobile node | Coastal areas | Tsunamic data | Hypo-center and magnitude | NOAA |
| [177] | UAV nodes | ODLOS | Aerial images data | Received frames/sec | Local drones |
| [185] | Smartphones | S-D environment | Acceleration data | Earthquake data | NIED and USGS |
| [188] | Seismometer | UG | GPS and weak motion data | Earthquake data | IRIS and NIED |
| [173] | MEMS | UG | Acceleration data | Acceleration, SNR | NIED |
| [174] | Arduino Cortex M4 | UG | Acceleration data | Earthquake detection accuracy and detection latency | Local data observed by MEMS accelerometers |
| [175] | Acceleration nodes | IDNLOS | Acceleration data | PGA and human activity | Local distributed smartphones |
| [176] | Soil and terrain nodes | UG | Soil moisture, shear strength of the soil, severity of the rain | Soil moisture, Soil shear strength, rain severity | GSI |
| [53] | Tmote Sky | ID and OD | Seismic velocity data | Location and magnitude | JMA and Hi-net |
| [179,195] | IoT gateway | UG | Seismic waveform | Earthquake predictions | Local datasets and regional data |
| [183] | Acceleration nodes | UG | Acceleration data | PGA | NIED |
| [185] | MEMS | Noisy environments | Seismic waveform | P-wave arrival | STEAD |
| [182] | Raspberry Pi | Mesh network | Seismic waveform | Local earthquake | Locally observed |
| [196] | SSN/SOSA ontology | UW | Volcanic data | Volcano-tectonic, long-period earthquakes, underwater explosions, and quarry blasts | Local data |

In order to prevent the loss of human life, the implementation of an EEWS is an absolute necessity. In order to effectively manage disasters and reduce the danger of earthquakes, it is essential to have the ability to promptly detect the features of an earthquake. With technologies already in place, such as the IoT network, social media, global positioning system (GPS), and mobile nodes, these attributes can be sent to help mitigate the effects of a catastrophic earthquake.

Figure 8 depicts a comprehensive EEWS with many administrations assisting in relieving the earthquake tragedy. The EEWS will include complete statistics regarding hospitals, railways, fire services, ambulances, airports, and so on based on these administrations. This proposed system does integrate social media, IoT technologies, cloud systems, and

mobile systems. It operates in two stages. The first stage is pre-disaster, as ML models are used to detect the commencement of the principal wave. This procedure is extremely advantageous for risk minimization, such as rapid shutting down of nuclear power plants, electrical producers, and so on. The second phase begins after the disaster has occurred, with the goal of mitigating/reducing the disaster's impacts. Using an integrated system, for example, allows for more accurate statistics regarding the affected people, buildings, utilities, and areas. As a result, an effective evacuation strategy can be implemented.

A solution that is both flexible and intelligent and that is able to deal with complex problems in a relatively short amount of time is required for such a system. ML has the potential to play a significant and critical part in the administrations that are interconnected and working on achieving successful EWS among the variety of existing current techniques. ML is a promising method that works regardless of the data type, format, length, and other factors such as these.

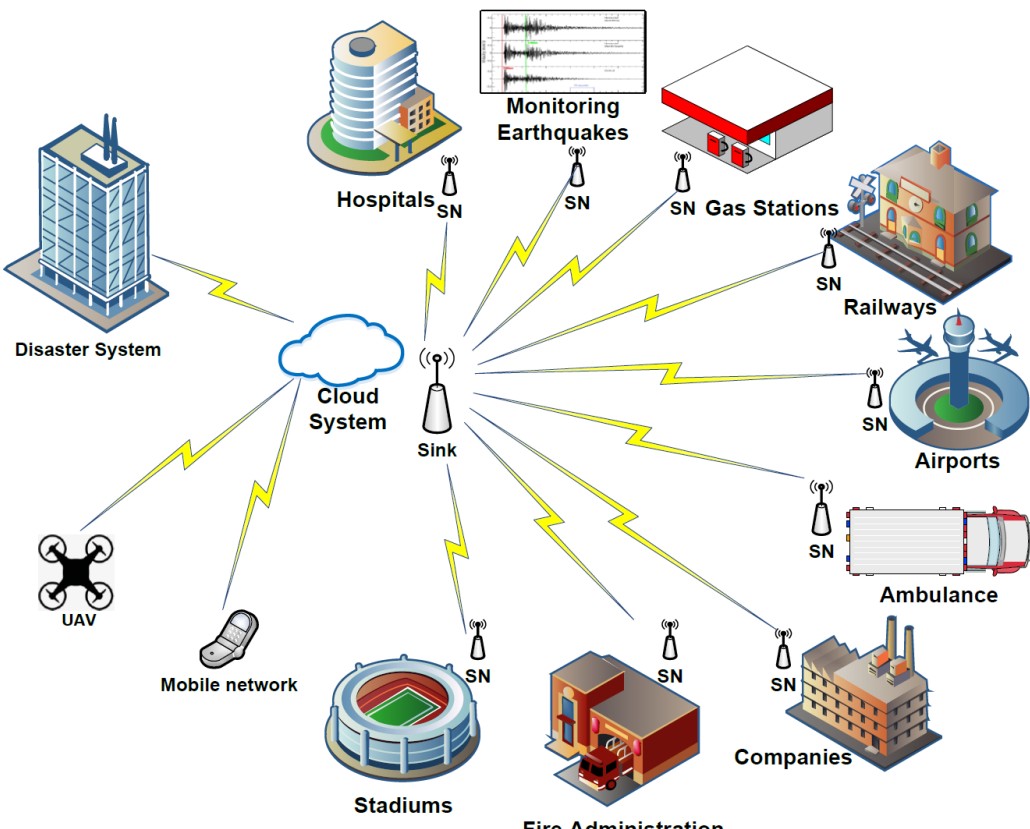

**Figure 8.** A general architecture of EWS.

Indeed, real-time monitoring takes place across all of the dispersed organizations shown in Figure 8, which serves as the foundation for a reliable EEWS. As a consequence of this, the transfer of data between various entities needs to be thoroughly investigated and estimated. After that, ML models are utilized to zero in on the current status of each object and even provide an estimate for a certain word. As a consequence of this, those institutions are capable of making useful contributions prior to, during, and after earthquake disasters. To put it another way, a technique such as this can assist with the management of earthquake catastrophes, the reduction of earthquake risks, and evacuation tactics. As a consequence of this, the performance of the EEWS improves in direct proportion to the quality of the ML model. Figure 9 provides a visual representation of the interaction between trains as a specific administration used in the process of full EEWS, the data processing, and the research done. To be more precise, earthquake data is monitored to be sent for processing using the IoT network in order to carry out the desired check and determine the correct decision to send to the railway system for suitable action using an ML model and the

railway information of the disaster location. This process is repeated until the appropriate decision is made.

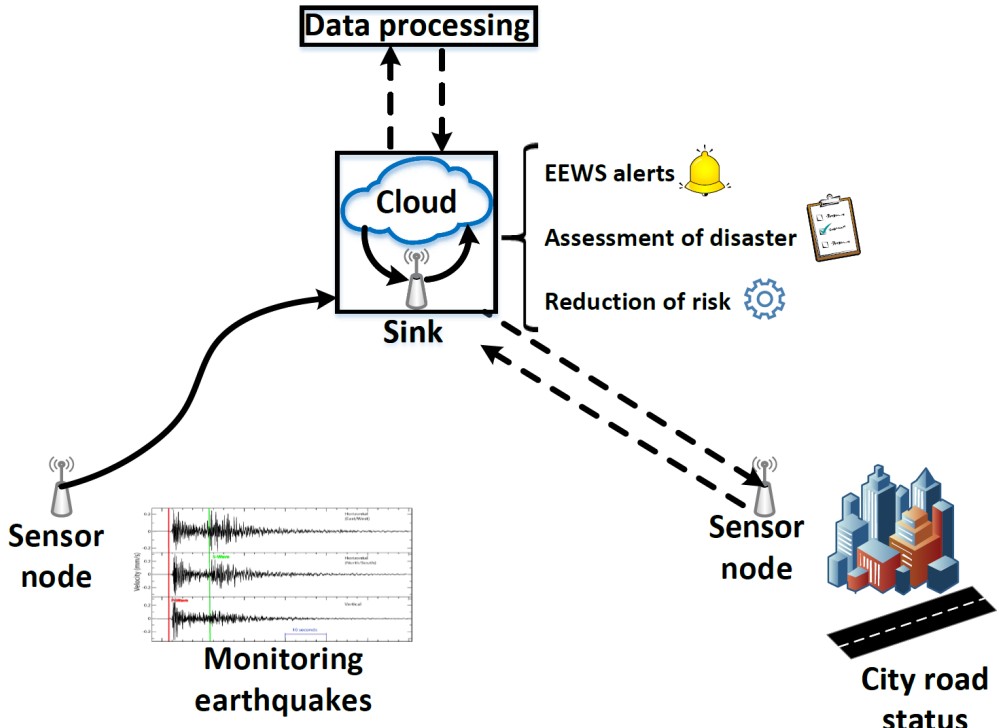

**Figure 9.** A pattern of Iot-based EEWS.

The authors of [197] mentioned the same approach of benefiting from UAVs as Aerial Base Stations (ABSs) to provide connectivity instead of traditional base stations. They proposed two trajectory planning algorithms using a k-value selection method and K-means centroids for UAVs. These UAVs served to the clusters of user equipment. By enhancing the study, the authors also gave two methods for cluster head selection to support continuous connectivity via UAVs and cluster heads.

The authors of [198] suggested a system to find the damage degrees of various earthquake region parts such as roads and riverways. They used single-rotor and six-rotor UAVs and took visible light images of the region parts. Once the image quality evaluation was done according to the image contrast, the image blur, and the image noise formulas, the aerial images were analyzed with Gray Level Cooccurrence Matrix, the Tamura, and the Gabor wavelet features. Lastly, the SVM classifier was used to obtain the damage levels.

The authors of [199] proposed to use UAVs for monitoring earthquake impacts after the disaster occurrence. The damage to the buildings was derived with the help of one fixed-wing UAV and two multirotor UAVs. The aerial mapping gathered from the UAVs was compared with a physical field survey. The buildings' structural properties were extracted from the damages on different parts of the surfaces, such as walls, roofs, and perimeter columns. Additionally, the liquefaction situation was seen from the area investigation that also presented the damage levels of the settlements.

Overall, the use of drones in earthquake disaster detection and management has the potential to save lives, speed up response times, and improve the efficiency of emergency services. As technology continues to advance, it is likely that drones will become an even more important tool in disaster management, helping to mitigate the effects of earthquakes and other natural disasters. Figure 10 shows the role of UAVs for three scenarios of pre, during, and post-disaster situations.

Evaluating the performance and reliability of IoT-enabled earthquake early warning systems (EEWS) is crucial for ensuring their effectiveness in real-world scenarios. There are several techniques that can be used to provide a comprehensive evaluation of the performance and reliability of these systems, including simulation testing, field testing, and data-driven analysis:

- Simulation testing involves creating a virtual environment that simulates real-world conditions, including seismic activity and sensor data [200,201]. Simulation testing allows researchers to test the performance of an EEWS system under different scenarios, such as different magnitudes and distances of earthquakes and different types of seismic waves [202]. This technique can also be used to evaluate the effectiveness of different algorithms and parameters used in the system [203].

- Field testing involves deploying an EEWS system in real-world conditions and collecting data on its performance and reliability [204,205]. Field testing can provide valuable insights into the system's performance under actual operating conditions, which may differ from those in a simulated environment. Field testing can also help to identify potential issues with the system, such as sensor malfunction or communication failures [206]. This technique can be time-consuming and resource-intensive, but it provides valuable data on the system's performance and reliability in real-world scenarios [207].

- Data-driven analysis involves analyzing large datasets generated by an EEWS system to identify patterns and trends, which can provide insights into its performance and reliability [208]. Data-driven analysis can be used to identify correlations between sensor data and earthquake characteristics, such as magnitude, duration, and intensity [209]. This technique can also be used to identify anomalies in sensor data, which may indicate issues with the system's performance or reliability [210]. Data-driven analysis can provide valuable insights into the performance and reliability of an EEWS system over long periods of time [211].

By using a combination of these techniques, researchers can gain a more comprehensive understanding of the performance and reliability of IoT-enabled EEWS systems. This can help to identify areas for improvement and ultimately improve the effectiveness of these systems in mitigating the impact of earthquakes.

Integrating advanced technologies such as ML algorithms, distributed computing, and edge computing into EEWS systems can improve their accuracy and effectiveness. However, there are several challenges and considerations associated with these technologies. For example, ML algorithms require large amounts of data and computational resources to train and optimize, which may be difficult to obtain in the context of EEWS systems [212,213]. Distributed computing can improve the scalability and fault tolerance of EEWS systems, but it also introduces additional complexity and overhead in terms of communication and coordination [214–217]. Edge computing can improve the responsiveness and efficiency of EEWS systems by processing data closer to the source. Still, it also requires careful management of resources and trade-offs between processing power and energy consumption [218,219]. In addition, the implementation of these advanced technologies can be complex and may require significant expertise and resources. Furthermore, there may be limitations associated with the hardware and software infrastructure of EEWS systems, such as sensor networks and communication protocols, which may need to be upgraded or modified to support these technologies. Therefore, while integrating advanced technologies into EEWS systems has the potential to improve their accuracy and effectiveness, careful consideration of the trade-offs and implementation complexities is necessary to ensure their successful deployment and operation.

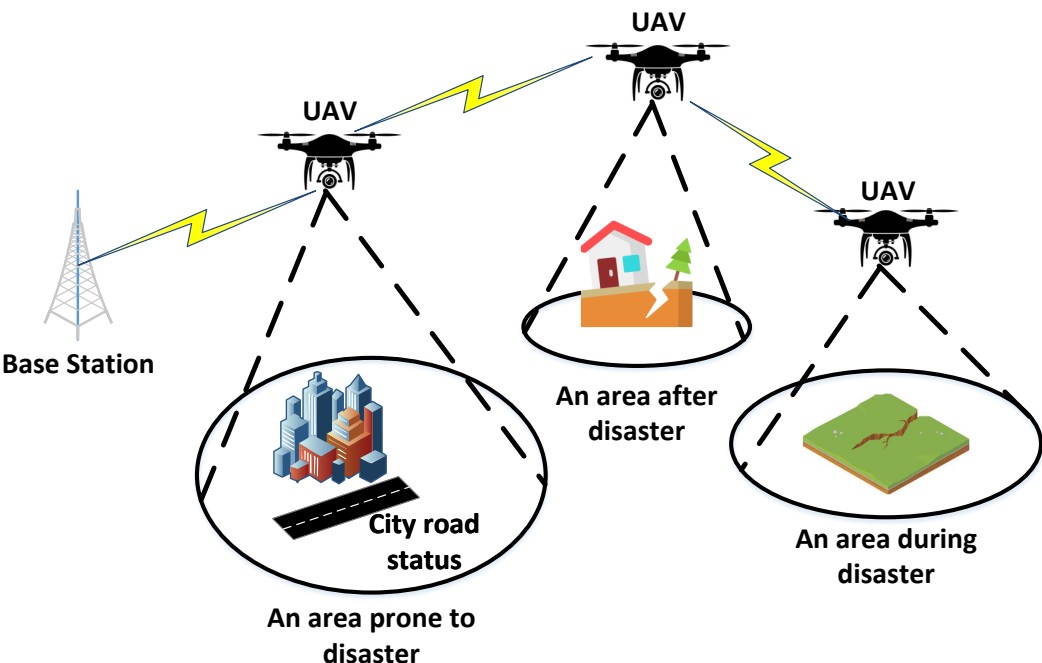

**Figure 10.** Three scenarios of disaster and the role of UAV.

## 5. Validation and Verification Aspects

Validation and verification (V&V) are essential for ensuring the quality, reliability, optimization, and safety of systems and products [220–222]. They involve rigorous evaluation against requirements and standards, identifying and correcting defects, improving performance, and ensuring compliance. In the context of IoT systems, V&V is crucial for addressing complex behaviors, identifying vulnerabilities, and complying with regulations [223,224]. For cloud systems, V&V ensures dependability, security, performance optimization, and adherence to standards [225–227]. Overall, V&V is vital for constructing trustworthy, resilient, compliant systems that meet user and societal expectations [228].

### 5.1. Different Categories of V&V Techniques

There are several types of validation and verification techniques that can be applied to these systems [229,230] (Figure 11). One common technique is functional testing, which involves testing the system's functions to ensure that they perform as expected [231]. This testing can be done manually or through automated testing tools. Another technique is performance testing, which involves testing the system's ability to handle a certain level of workload or traffic [232]. This can include stress testing, load testing, and capacity testing [233,234].

Security testing is another important technique for IoT and cloud systems [235–237]. This involves testing the system's security features and protocols to identify vulnerabilities and ensure that sensitive information is protected [238–240]. Penetration testing is a type of security testing that involves attempting to hack into the system to identify weaknesses and potential security breaches [241–243].

Usability testing is also important for these systems, as they must be user-friendly and easy to navigate [244]. This testing involves gathering feedback from users to identify areas of the system that can be improved to enhance the user experience [245].

Regression testing is another important technique for IoT and cloud systems [246,247]. It involves retesting the system after making changes or updates to ensure that the changes have not introduced new defects or issues in previously tested areas [248]. Regression testing can be done manually or through automated testing tools, and it is important to perform regularly to ensure that the system remains stable and reliable throughout its

lifecycle [249,250]. By conducting thorough regression testing, developers can identify and fix issues early on, ultimately leading to a more robust and reliable system for users [251].

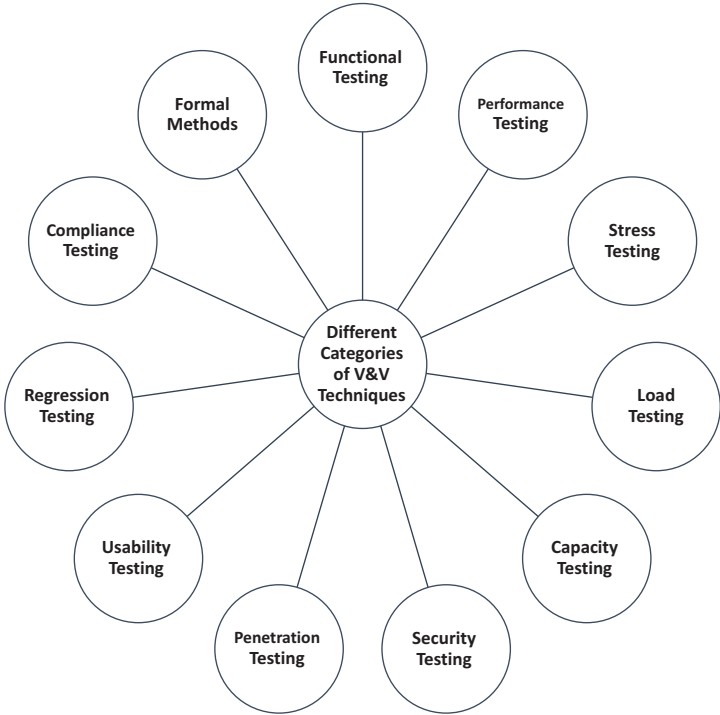

**Figure 11.** Different Categories of V&V Techniques.

Formal methods (Figure 12) are another important set of techniques that can be used for validation and verification of IoT and cloud systems [252,253]. Formal methods involve the use of mathematical models and logic to verify the correctness and reliability of a system [254]. This technique can be used to check the consistency of the system design and its specifications, as well as to identify potential errors and defects in the system [255,256]. Formal methods can also be used to ensure that the system meets certain performance and safety requirements [257–259]. While formal methods can be more time-consuming and complex than other validation and verification techniques, they can provide a high level of confidence in the correctness and reliability of the system [260–262]. Therefore, formal methods are an important tool for developers to consider when designing and testing IoT and cloud systems.

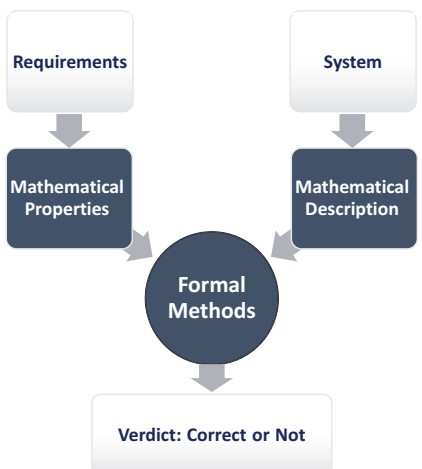

**Figure 12.** A simplified illustration of how Formals Methods work [263].

In addition to these techniques, there are also validation and verification techniques specific to cloud systems. For example, validation techniques for cloud systems may include compliance testing to ensure that the system complies with industry standards and regulations and testing the system's ability to handle real-time data processing and communication.

Overall, validation and verification techniques are critical for ensuring the reliability, security, and efficiency of IoT and cloud systems. By applying these techniques, developers can identify and address issues early in the development process, ultimately leading to better-performing and more secure systems for users.

### 5.2. Adaptation of V&V Techniques for EEWS

Verification and validation (V&V) techniques are crucial for ensuring the reliability and effectiveness of earthquake detection and warning systems in the context of IoT. These systems use a combination of sensors, data analysis algorithms, and communication technologies to detect and respond to earthquakes in real time. However, the adaptation of V&V techniques for these systems presents unique challenges and opportunities.

One challenge is the need for a robust and reliable sensor network. EEWS relies on a sensor network to detect and measure seismic activity. These sensors must be calibrated and tested regularly to ensure that they are functioning correctly. V&V techniques can help to ensure that the sensor network is reliable and accurate by providing a framework for testing and calibration. More precisely, to ensure the reliability and accuracy of the entire sensor network, a statistical calibration and testing approach can be employed. This involves validating a representative sample of sensors rather than individually testing each sensor, given the large number of sensors typically used. Statistically, the majority of sensors will be sufficiently calibrated and functioning properly at the time of an earthquake.

Another challenge is the desirability of real-time data analysis and action plans. In earthquake detection systems, timely and accurate decision-making is critical for minimizing damage and saving lives. V&V techniques can help to ensure that the data analysis algorithms used in these systems are accurate and reliable. This can involve testing the algorithms under a variety of conditions and scenarios, as well as ensuring that they are able to operate in real time.

In addition to challenges, there are also opportunities for the adaptation of V&V techniques for IoT earthquake systems. One opportunity is the use of simulation and modeling. V&V techniques can be used to create simulations and models of earthquake scenarios to test and validate the performance of the detection and warning systems. This can help to identify potential weaknesses in the system and inform improvements.

Another opportunity is the use of crowdsourcing and citizen science. V&V techniques can be used to validate and integrate data from citizen scientists and volunteers who contribute to earthquake detection and warning systems. This can help to improve the accuracy and reliability of the system while also engaging the public in the process.

In conclusion, the adaptation of V&V techniques for IoT earthquake systems presents both challenges and opportunities. By leveraging V&V techniques, developers can ensure that these systems are reliable, accurate, and effective in detecting and responding to seismic activity. This can help to minimize damage and save lives in the event of an earthquake.

### 5.3. Cost and Limitations of V&V Techniques

Verification and validation (V&V) techniques are crucial in ensuring the quality and reliability of software systems for IoT and cloud computing systems. However, these systems come with unique challenges and limitations that must be considered when implementing V&V techniques.

One cost of V&V techniques for IoT and cloud systems is the sheer scale of these systems. These systems can involve thousands or even millions of interconnected devices, making it difficult to perform comprehensive testing and validation. Additionally, the het-

erogeneity of these systems, with devices from different manufacturers and with different capabilities, can complicate the verification and validation process.

Another cost of V&V techniques for IoT and cloud systems is the need for specialized testing environments and tools. These systems require highly specialized tools and environments for testing and validation, such as simulators, emulators, and testbeds. These tools can be expensive to implement and maintain and may require highly skilled personnel to operate effectively.

Furthermore, there are limitations to the effectiveness of V&V techniques for IoT and cloud systems. One limitation is the difficulty of testing for security and privacy vulnerabilities. These systems often involve sensitive data and critical infrastructure, making security and privacy a top priority. However, it is challenging to test for all possible security and privacy vulnerabilities, especially as new threats emerge constantly.

Another limitation is the challenge of testing for real-time and low-latency requirements. IoT and cloud systems often require real-time performance and low-latency communication, which can be difficult to test and validate. These systems may involve complex interactions between devices and services, making it challenging to ensure that they meet these requirements.

In conclusion, V&V techniques are crucial for ensuring the quality and reliability of software systems for IoT and cloud computing systems. However, these systems have unique challenges and limitations that must be considered when implementing V&V techniques. By understanding these costs and limitations, software developers can implement V&V techniques more effectively and efficiently, ultimately leading to higher-quality software systems.

## 6. Open Challenges, Conclusions and Future Directions

The use of IoT and cloud facilities for EEWS presents significant opportunities for improving the speed and accuracy of earthquake detection and response. However, there are also several challenges that must be addressed to ensure the reliability and effectiveness of these systems (Figure 13).

- Sensor network reliability and accuracy: One of the primary challenges in implementing IoT and cloud-based EEWS is ensuring the reliability and accuracy of the sensor network. These systems rely on a network of sensors to detect and measure seismic activity, making it essential to ensure that the sensors are functioning correctly.
- Real-time data processing and decision-making: EEWS require fast and accurate data processing and decision-making capabilities to provide timely alerts to people and organizations in affected areas. This requires sophisticated algorithms and real-time data processing capabilities, which can be challenging to implement in IoT and cloud-based systems.
- Secure communication channels: The transmission of data between sensors, cloud facilities, and other components in an EEWS must be secure to prevent unauthorized access and tampering. Ensuring the security of communication channels is a significant challenge in designing and implementing these systems.
- Heterogeneity and scalability: IoT and cloud-based systems are inherently heterogeneous, with devices and services from different manufacturers and with different capabilities. Ensuring seamless integration and scalability of these systems is a significant challenge, particularly as the number of devices and sensors in the network increases.
- Cost-effectiveness and sustainability: Implementing an EEWS using IoT and cloud facilities can be costly, requiring significant investment in hardware, software, and personnel. Ensuring the cost-effectiveness and sustainability of these systems is a significant challenge, particularly in regions with limited resources.
- Usability and accessibility: EEWS must be usable and accessible to people and organizations in affected areas, including those with limited literacy or technical skills. Ensuring the usability and accessibility of these systems is a significant challenge, requiring careful consideration of user needs and preferences.

- Privacy and ethical concerns: The collection and processing of data in EEWS raise privacy and ethical concerns, particularly as these systems become more sophisticated and widespread. Ensuring that these systems comply with relevant regulations and ethical principles is a significant challenge.
- Interference from environmental factors: EEWS can be affected by environmental factors such as electromagnetic noise and weather conditions, which can interfere with the accuracy and reliability of the sensor network. Ensuring the robustness and resilience of these systems is a significant challenge, requiring careful consideration of environmental factors.
- Continuous monitoring and maintenance: IoT and cloud-based EEWS require continuous monitoring and maintenance to ensure system performance and reliability. Ensuring the continuous monitoring and maintenance of these systems is a significant challenge, requiring robust and scalable infrastructure and skilled personnel.

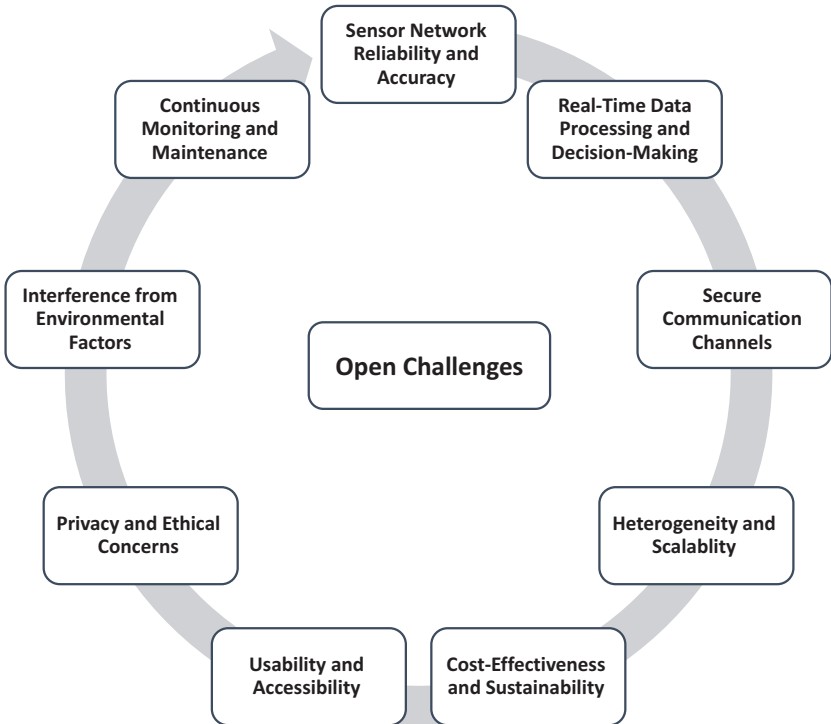

**Figure 13.** Open Challenges.

The use of IoT and cloud facilities for EEWS presents a significant opportunity for improving the speed and accuracy of earthquake detection and response. However, addressing the challenges outlined above is essential to ensure the reliability and sustainability of these systems in the long term. This survey has highlighted the potential benefits of using IoT and cloud technologies in EEWS, including real-time data analysis, improved sensor networks, and faster decision-making. However, the survey has also identified several challenges that must be addressed, such as the need for reliable and accurate sensor networks, real-time data processing, and secure communication channels. Overall, the survey underscores the importance of continued research and development in this area, as well as the need for rigorous verification and validation techniques to ensure the reliability and effectiveness of these systems. Below, we propose some interesting future directions:

1. Development of more efficient and accurate sensors: Research and development should focus on developing more efficient and accurate sensors that can accurately detect and measure seismic activity while also being cost-effective and scalable.
2. Integration of artificial intelligence (AI) and ML: The integration of AI and ML can help to improve the accuracy and reliability of data analysis algorithms used in EEWS.

This can lead to faster and more accurate decision-making, improving the effectiveness of these systems [264,265].

3. Standardization of communication protocols: The standardization of communication protocols can help to ensure the interoperability and scalability of IoT and cloud-based EEWS. This can simplify the integration of different devices and services, reducing the complexity of these systems.

4. Adoption of free, open-source software: The adoption of free, open-source software can help to reduce the cost and complexity of developing EEWS while also encouraging collaboration and innovation in this area.

5. Engagement with local communities: Engagement with local communities can help to ensure that EEWS are developed in a form that meets the needs and preferences of people and organizations in affected areas. This can improve the usability and effectiveness of these systems in real-world scenarios.

6. Development of new funding models: The development of new funding models, such as public–private partnerships, can help to ensure the sustainability and scalability of EEWS. This can provide the necessary resources and expertise to develop and maintain these systems over the long term.

7. The "last kilometer" problem: This problem is the difficulty of assuring prompt and efficient warning, communication, and reaction systems to people and communities in the final seconds before the occurrence of powerful and devastating S-wave shaking during an earthquake. In particular, it requires addressing densely populated areas where the window for preparation and evacuation is constrained, where there is a gap between earthquake EEWS and the capacity to reach and notify individuals in the impacted area. In order to protect people's safety and well-being in the final crucial seconds before the arrival of the destructive seismic waves, this topic focuses on the necessity for the effective broadcast of alerts and emergency instructions.

In conclusion, continued research and development, as well as collaboration and innovation, will be essential in addressing the challenges and realizing the potential benefits of using IoT and cloud facilities for EEWS. By addressing these challenges and implementing future directions, it is possible to develop more reliable, accurate, and effective EEWS that can save lives and minimize damage in the event of seismic activity.

**Author Contributions:** Conceptualization, M.S.A. and M.K.; methodology, M.S.A., M.K. and D.Y.-K.; investigation, M.S.A., M.K. and I.B.D.; writing—original draft preparation, M.S.A., M.K. and D.Y.-K.; writing—review and editing, M.S.A., M.K., I.B.D. and W.Y.H.A.; supervision, M.S.A. and M.K.; resources, M.S.A., M.K. and I.B.D.; data curation, M.S.A. and M.K.; visualization, M.S.A. and M.K. All authors have read and agreed to the published version of the manuscript.

**Funding:** This research received no external funding.

**Institutional Review Board Statement:** Not applicable.

**Informed Consent Statement:** Not applicable.

**Data Availability Statement:** Not applicable.

**Conflicts of Interest:** The authors declare no conflict of interest.

## Abbreviations

| | |
|---|---|
| EEWS | Earthquake Early Warning Systems |
| SDN | Software Defined Network |
| AI | Artificial Intelligence |
| NFV | Network Functions Virtualization |
| DMSEEW | Distributed Multi-Sensor Earthquake Early Warning |
| Micro-MEMS | Micro-Electro-Mechanical systems |
| ML | Machine Learning |
| IoT | Internet of Things |

| UG | Underground |
| ODLOS | Outdoor Line-of-sight |
| UAV | Unmanned Arial Vehicle |
| IDLOS | Indoor Line-of-sight |
| UW | Under Water |
| OD | Outdoor |
| ID | Indoor |
| DT | Decision Tree |
| RF | Random Forest |
| SVM | Support Vector Machine |
| NB | Naïve Bayes |
| KNN | K-Nearest Neighbor |
| FD | Federated Learning |
| GPS | Global Positioning System |
| 5G | Fifth Generation |
| B5G | Beyond Fifth Generation |
| AE | Autoencoder |
| CNN | Convolutional Neural Network |
| Body waves | P/S-wave |
| NIED | National Research Institute of Earth Science and Disaster |
| V&V | Verification and Verification |

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
