# Peer review of "Early Detection of Earthquakes Using IoT and Cloud Infrastructure: A Survey"

_sustainability, doi:10.3390/su151511713_

Round 1
Reviewer 1 Report
The paper would benefit from providing more in-depth technical details on the signal processing techniques employed for earthquake detection and analysis. Explaining the specific algorithms and methodologies used for seismic wave analysis would enhance the paper's technical rigor.
It would be helpful to provide a comprehensive evaluation of the IoT-enabled EEWS system's performance and reliability. Including quantitative metrics and results from experiments or simulations would strengthen the validity and credibility of the proposed system.
The paper could explore the challenges and considerations associated with integrating advanced technologies such as machine learning algorithms, distributed computing, and edge computing into the EEWS system. Discussing the potential limitations, trade-offs, and implementation complexities would enhance the paper's technical depth.
Providing a detailed description of the generic EEWS architecture would be beneficial. Explaining the components, their interactions, and the underlying infrastructure would allow readers to understand the system's structure and functionality in more detail.
The paper could further elaborate on the role of drones in disaster management and their specific applications in enhancing the effectiveness of EEWS. Discussing the technical aspects, such as drone-based sensing and data collection, real-time communication, and coordination with the IoT and cloud infrastructure, would enrich the technical content of the paper.
Considering the significance of verification and validation in EEWS, the paper should provide more insights into the specific methods used for evaluating and validating the proposed system. Explaining the experimental setups, data sources, and criteria for performance assessment would enhance the technical robustness of the study.
Kindly refrain from using sources that were released before 2019. Cite recent studies that are highly relevant to your subject. The paper also doesn't have enough citations. Another key stage is to compare the topic of the article to other recent publications or works that are comparable to broaden the research's ramifications beyond the subject. Authors may cite and rely on these important works while discussing the subject of their article and the problems of the present.
A. Chen, J., Wen, L., Bi, C., Liu, Z., Liu, X., Yin, L.,... Zheng, W. (2023). Multifractal analysis of temporal and spatial characteristics of earthquakes in Eurasian seismic belt. Open Geosciences, 15(1). doi: doi:10.1515/geo-2022-0482
B. Heidari, N. J. Navimipour, M. A. J. Jamali, and S. Akbarpour, "A green, secure, and deep intelligent method for dynamic IoT-edge-cloud offloading scenarios," Sustainable Computing: Informatics and Systems, vol. 38, p. 100859, 2023.
C. Chen, P., Liu, H., Xin, R., Carval, T., Zhao, J., Xia, Y.,... Zhao, Z. (2022). Effectively Detecting Operational Anomalies In Large-Scale IoT Data Infrastructures By Using A GAN-Based Predictive Model. The Computer Journal, 65(11), 2909-2925. doi: 10.1093/comjnl/bxac085
D. Mehdi Darbandi; “Proposing New Intelligence Algorithm for Suggesting Better Services to Cloud Users based on Kalman Filtering”; Published by Journal of Computer Sciences and Applications (ISSN: 2328-7268), Vol. 5, Issue 1, 2017; PP. 11-16; DOI: 10.12691/JCSA-5-1-2; USA.
E. Dang, P., Cui, J., Liu, Q., & Li, Y. (2023). Influence of source uncertainty on stochastic ground motion simulation: a case study of the 2022 Mw 6.6 Luding, China, earthquake. Stochastic Environmental Research and Risk Assessment. doi: 10.1007/s00477-023-02427-y
Moderate editing of English language required
Author Response
Reply to the Editor and Reviewers’ Comments
Paper ID: sustainability-2472029
Paper title: Sustainable Early Detection of Earthquakes using IoT and Cloud Infrastructure: A Survey
We would like to thank the editor and the reviewers for their valuable comments on the paper (sustainability-2472029) entitled “Sustainable Early Detection of Earthquakes using IoT and Cloud Infrastructure: A Survey’’. We have revised the manuscript according to the reviewers' comments which helped in improving the quality and presentation of the paper. In order to facilitate our reply, the amendments are clearly with red color with highlights in the revised manuscript to clarify them. We hope that the revised version has addressed the reviewers' comments. Our detailed reply to the comments we have received is given next.
Reviewer 1 comments
1- The paper would benefit from providing more in-depth technical details on the signal processing techniques employed for earthquake detection and analysis. Explaining the specific algorithms and methodologies used for seismic wave analysis would enhance the paper's technical rigor. |
Reply: We want to express our gratitude to the reviewer for their thorough work, revisions, and recommendations. We have carefully considered the reviewer’s comments. We added the following paragraphs to the paper: "Seismic wave analysis is a key component of earthquake early warning systems, as it enables the detection and characterization of seismic waves in real-time [107,108]. One of the most widely used signal processing techniques in seismic wave analysis is the Fourier transform, which is used to transform time-domain signals into frequency-domain signals [109,110]. In earthquake early warning systems, the Fourier transform is often used to analyze the spectral content of seismic waves, which can provide important information about the location, magnitude, and duration of an earthquake [111]. The Fourier transform is also used to filter out noise and unwanted signals from seismic data, improving the accuracy of earthquake detection and analysis [112]. Another advanced signal processing technique used in seismic wave analysis is wavelet analysis, which is used to analyze signals that are both time-varying and non-stationary [113–115]. In earthquake early warning systems, wavelet analysis is often used to detect and analyze seismic waves that have complex frequency components, such as those generated by slow earthquakes or volcanic activity [116,117]. By decomposing a seismic waveform into its constituent frequency components, wavelet analysis can provide more detailed information on the characteristics of seismic waves, such as their frequency content, duration, and amplitude [118,119]. In addition to these advanced signal processing techniques, earthquake early warning systems also rely on a variety of specific parameters to optimize their performance [120,121]. These parameters include sampling rates, window sizes, and filter cutoff frequencies, among others. Sampling rates determine how often seismic data is collected and stored, while window sizes determine the length of time over which seismic data is analyzed. Filter cutoff frequencies determine which frequency components of a seismic waveform are analyzed, and are often used to remove noise and unwanted signals from seismic data. In conclusion, earthquake early warning systems rely on a variety of advanced signal processing techniques and specific parameters to detect and analyze seismic waves in real-time. By providing more detailed information on these techniques and parameters, we aim to enhance the technical rigor of our paper and improve the understanding of the underlying technology. By optimizing the performance of earthquake early warning systems through advanced signal processing techniques and specific parameters, we can improve the accuracy and effectiveness of these systems, ultimately helping to save lives and reduce the impact of earthquakes on communities." |
2- It would be helpful to provide a comprehensive evaluation of the IoT-enabled EEWS system's performance and reliability. Including quantitative metrics and results from experiments or simulations would strengthen the validity and credibility of the proposed system. |
Reply: We thank the reviewer for the comment. We added the following paragraphs to the paper: " Evaluating the performance and reliability of IoT-enabled earthquake early warning systems (EEWS) is crucial for ensuring their effectiveness in real-world scenarios. There are several techniques that can be used to provide a comprehensive evaluation of the performance and reliability of these systems, including simulation testing, field testing, and data-driven analysis: · Simulation testing involves creating a virtual environment that simulates real-world conditions, including seismic activity and sensor data [200,201]. Simulation testing allows researchers to test the performance of an EEWS system under different scenarios, such as different magnitudes and distances of earthquakes, and different types of seismic waves [202]. This technique can also be used to evaluate the effectiveness of different algorithms and parameters used in the system [203]. · Field testing involves deploying an EEWS system in real-world conditions and collecting data on its performance and reliability [204,205]. Field testing can provide valuable insights into the system's performance under actual operating conditions, which may differ from those in a simulated environment. Field testing can also help to identify potential issues with the system, such as sensor malfunction or communication failures [206]. This technique can be time-consuming and resource-intensive, but it provides valuable data on the system's performance and reliability in real-world scenarios [207]. · Data-driven analysis involves analyzing large datasets generated by an EEWS system to identify patterns and trends, which can provide insights into its performance and reliability [208]. Data-driven analysis can be used to identify correlations between sensor data and earthquake characteristics, such as magnitude, duration, and intensity [209]. This technique can also be used to identify anomalies in sensor data, which may indicate issues with the system's performance or reliability [210]. Data-driven analysis can provide valuable insights into the performance and reliability of an EEWS system over long periods of time [211]. By using a combination of these techniques, researchers can gain a more comprehensive understanding of the performance and reliability of IoT-enabled EEWS systems. This can help to identify areas for improvement and ultimately improve the effectiveness of these systems in mitigating the impact of earthquakes." |
3- The paper could explore the challenges and considerations associated with integrating advanced technologies such as machine learning algorithms, distributed computing, and edge computing into the EEWS system. Discussing the potential limitations, trade-offs, and implementation complexities would enhance the paper's technical depth. |
Reply: We thank the reviewer for the comment. We added the following paragraph to the paper: " Integrating advanced technologies such as machine learning algorithms, distributed computing, and edge computing into EEWS systems can improve their accuracy and effectiveness. However, there are several challenges and considerations associated with these technologies. For example, machine learning algorithms require large amounts of data and computational resources to train and optimize, which may be difficult to obtain in the context of EEWS systems [212,213]. Distributed computing can improve the scalability and fault tolerance of EEWS systems, but it also introduces additional complexity and overhead in terms of communication and coordination [214–217]. Edge computing can improve the responsiveness and efficiency of EEWS systems by processing data closer to the source, but it also requires careful management of resources and trade-offs between processing power and energy consumption [218,219]. In addition, the implementation of these advanced technologies can be complex and may require significant expertise and resources. Furthermore, there may be limitations associated with the hardware and software infrastructure of EEWS systems, such as sensor networks and communication protocols, which may need to be upgraded or modified to support these technologies. Therefore, while integrating advanced technologies into EEWS systems has the potential to improve their accuracy and effectiveness, careful consideration of the trade-offs and implementation complexities is necessary to ensure their successful deployment and operation. " |
4- Providing a detailed description of the generic EEWS architecture would be beneficial. Explaining the components, their interactions, and the underlying infrastructure would allow readers to understand the system's structure and functionality in more detail. |
Reply: We thank the reviewer for the comment. We have added the following paragraph: " A generic EEWS architecture typically consists of three main components: the seismic network, the processing center, and the alert distribution system [165]. The seismic network is composed of a set of sensors deployed across a region of interest, which detect and record seismic waves generated by earthquakes. The sensor data is transmitted to the processing center, where it is analyzed in real time using algorithms and models to estimate the location, magnitude, and other characteristics of the earthquake [166]. The alert distribution system then disseminates the earthquake alert to end-users through various channels, such as mobile devices, sirens, and public announcements [167]. The underlying infrastructure of the EEWS includes a variety of hardware and software components, including seismometers, communication networks, computing systems, and databases [168]. The seismometers are typically deployed in a dense network to ensure high spatial resolution and coverage, and they are connected to a communication network that transmits the sensor data to the processing center [169]. The processing center is composed of a set of computing systems that perform real-time data analysis, using a variety of algorithms and models to estimate the earthquake parameters [170]. The alert distribution system includes a set of communication channels and protocols that disseminate the alert to end-users, as well as a database that stores historical and real-time earthquake data [171]. The interactions between these components are tightly coordinated to ensure timely and accurate earthquake alerts, which can help to mitigate the impact of earthquakes and save lives [172]. " |
5- The paper could further elaborate on the role of drones in disaster management and their specific applications in enhancing the effectiveness of EEWS. Discussing the technical aspects, such as drone-based sensing and data collection, real-time communication, and coordination with the IoT and cloud infrastructure, would enrich the technical content of the paper. |
Reply: We thank the reviewer for the comment. We added the following paragraph: " In recent years, advances in technology have enabled the integration of new sensing and data collection methods into EEWS systems, including the use of drones [151,152]. Drone-based sensing can provide high-resolution data on earthquake characteristics, such as ground motion and deformation, which can improve the accuracy and effectiveness of EEWS systems [153,154]. Real-time communication is also essential for the timely dissemination of earthquake alerts, and various communication technologies, including satellite and wireless networks, are used to transmit sensor data and alerts to processing centers and end-users [155]. In addition, the integration of EEWS systems with IoT and cloud infrastructure can provide scalability, fault-tolerance, and data processing capabilities [156]. IoT devices, such as accelerometers and GPS sensors, can provide additional data sources for earthquake monitoring and analysis, while cloud infrastructure can provide storage and processing capabilities for large-scale data analysis and modeling [157]. Coordination between these various technical aspects is essential for the successful deployment and operation of EEWS systems, and careful consideration of their capabilities, limitations, and interdependencies is necessary to ensure their effectiveness in mitigating the impact of earthquakes [158]. " |
6- Considering the significance of verification and validation in EEWS, the paper should provide more insights into the specific methods used for evaluating and validating the proposed system. Explaining the experimental setups, data sources, and criteria for performance assessment would enhance the technical robustness of the study. |
Reply: We thank the reviewer for the comment. We answered this point in our response to point 2. |
7- Kindly refrain from using sources that were released before 2019. Cite recent studies that are highly relevant to your subject. The paper also doesn't have enough citations. Another key stage is to compare the topic of the article to other recent publications or works that are comparable to broaden the research's ramifications beyond the subject. Authors may cite and rely on these important works while discussing the subject of their article and the problems of the present. A. Chen, J., Wen, L., Bi, C., Liu, Z., Liu, X., Yin, L.,... Zheng, W. (2023). Multifractal analysis of temporal and spatial characteristics of earthquakes in Eurasian seismic belt. Open Geosciences, 15(1). doi: doi:10.1515/geo-2022-0482 B. Heidari, N. J. Navimipour, M. A. J. Jamali, and S. Akbarpour, "A green, secure, and deep intelligent method for dynamic IoT-edge-cloud offloading scenarios," Sustainable Computing: Informatics and Systems, vol. 38, p. 100859, 2023. C. Chen, P., Liu, H., Xin, R., Carval, T., Zhao, J., Xia, Y.,... Zhao, Z. (2022). Effectively Detecting Operational Anomalies In Large-Scale IoT Data Infrastructures By Using A GAN-Based Predictive Model. The Computer Journal, 65(11), 2909-2925. doi: 10.1093/comjnl/bxac085 D. Mehdi Darbandi; “Proposing New Intelligence Algorithm for Suggesting Better Services to Cloud Users based on Kalman Filtering”; Published by Journal of Computer Sciences and Applications (ISSN: 2328-7268), Vol. 5, Issue 1, 2017; PP. 11-16; DOI: 10.12691/JCSA-5-1-2; USA. E. Dang, P., Cui, J., Liu, Q., & Li, Y. (2023). Influence of source uncertainty on stochastic ground motion simulation: a case study of the 2022 Mw 6.6 Luding, China, earthquake. Stochastic Environmental Research and Risk Assessment. doi: 10.1007/s00477-023-02427-y |
Reply: We thank the reviewer for the comment. We included all the references proposed by the reviewer along with other recent interesting references. |

Reviewer 2 Report
The paper is a survey on IoT usage for earthquake early warning systems. Being a survey, it does not have a very strong novelty other than organizing the presentation of ideas. It does have merit doing so, but in my opinion it lacks the aspect of how IoT, 5G and B5G can be used to provide emergency connectivity, including aspects such as D2D communication. The points on V&V presented in the paper are not the same because they cover different aspects of the problem. The authors explore how drones may be employed but do not explore edge computing possibilities. Including these aspects are not a requirement but would benefit the work.
English is fine except for some minor spelling and grammar errors. However, there are times in which the ideas are not well connected. For example, in the introduction, beginning in line 67, the paragraph presents some ideas that begin with communication aspects but ends with robot-event.
Also, the way references are made is not standardized. E.g., when citing [9], the author is cited by name. Sometimes the work is referenced using only the number without mentioning the authors whatsoever (e.g. when citing [1]*), or mentioning the authors but not their names (e.g. when citing [115]). Please verify the standard.
Author Response
Reply to the Editor and Reviewers’ Comments
Paper ID: sustainability-2472029
Paper title: Sustainable Early Detection of Earthquakes using IoT and Cloud Infrastructure: A Survey
We would like to thank the editor and the reviewers for their valuable comments on the paper (sustainability-2472029) entitled “Sustainable Early Detection of Earthquakes using IoT and Cloud Infrastructure: A Survey’’. We have revised the manuscript according to the reviewers' comments which helped in improving the quality and presentation of the paper. In order to facilitate our reply, the amendments are clearly with red color with highlights in the revised manuscript to clarify them. We hope that the revised version has addressed the reviewers' comments. Our detailed reply to the comments we have received is given next.
Reviewer 2 comments
1- The paper is a survey on IoT usage for earthquake early warning systems. Being a survey, it does not have a very strong novelty other than organizing the presentation of ideas. It does have merit doing so, but in my opinion it lacks the aspect of how IoT, 5G and B5G can be used to provide emergency connectivity, including aspects such as D2D communication. The points on V&V presented in the paper are not the same because they cover different aspects of the problem. The authors explore how drones may be employed but do not explore edge computing possibilities. Including these aspects are not a requirement but would benefit the work. |
Reply: We appreciate your insight into the important points that we overlooked in our initial submission and for bringing them to our attention. In response to your comments, we have added some additional paragraphs to our paper to address the potential of 5G and B5G networks for emergency connectivity, including aspects such as D2D communication and edge computing. We acknowledge that these aspects are crucial in enhancing the capabilities of earthquake early warning systems, and we appreciate your suggestions in this regard. While we have included some discussion on these points in our paper, we agree that they require more in-depth analysis and consideration. We are committed to exploring these aspects further in our future works and will incorporate them into our research agenda going forward. The added paragraphs (in the introduction) are the following: "5G and B5G networks offer several advantages for emergency communication, including faster data transmission speeds, lower latency, and improved reliability [57–59]. These networks can enable real-time communication between emergency responders and affected individuals, as well as the seamless transfer of data and video feeds from IoT devices, such as sensors and drones [60–62]. In particular, D2D communication can play a crucial role in emergency situations, as it allows devices to communicate directly with each other without relying on a centralized network [63,64]. This can be especially useful in scenarios where network infrastructure may be damaged or overloaded, as D2D communication can operate on a peer-to-peer basis and bypass the need for a central network [65,66]. In an earthquake early warning system, D2D communication could allow sensors to share data with each other and trigger alerts in real-time, without relying on a centralized system [67,68]. Edge computing can also be leveraged to enhance the performance of earthquake early warning systems [69,70]. By processing data closer to the source, edge computing can reduce the amount of data that needs to be transmitted to centralized servers and enable faster response times [71,72]. For example, in an earthquake early warning system that uses drones to collect data, edge computing could be used to process the data on the drones themselves, rather than transmitting it back to a central server for processing [73,74]. This would not only reduce the amount of data that needs to be transmitted, but also enable faster response times in the event of an earthquake [75]. "
|
2- English is fine except for some minor spelling and grammar errors. However, there are times in which the ideas are not well connected. For example, in the introduction, beginning in line 67, the paragraph presents some ideas that begin with communication aspects but ends with robot-event. |
Reply: We thank the reviewer for the comment. We have done a thoroughly revision for the manuscript to avoid any spelling and grammar error. We also appreciate the second comment of the reviewer. We did our best for solving this issue. For the mentioned example, we divided the paragraph into two paragraphs and we added content to each part in order to make the ideas clearer for readers. Here are the two new paragraphs: Radio-frequency identification, satellite systems, the Internet of Things (IoT), network functions virtualization (NFV), 5G, software-defined networks (SDN), data networks, and a variety of other technologies have all been the focus of significant research in recent years in an effort to lessen the damage that earthquakes cause [ 24– 32]. For instance, satellite systems have been used to track earthquake movements, and IoT sensors have been used to detect earthquakes and provide early warnings. Furthermore, 5G and SDN technologies have been deployed for real-time communication and data transmission in emergency situations. These technologies have greatly enhanced the76 accuracy and speed of earthquake detection and warning systems, and have improved the response time of emergency services. Moreover, the integration of robots and the internet has the potential to be a significant breakthrough in this field. According to [33], a new integrated system named "robot-event" has been proposed, which is able to execute autonomous inspections and emergency responses to a severe event. The robot uses real-time image tracking to inspect the indoor environment and help any human victims found on the ground. It operates in structurally sound houses with moderate damage, focusing on situations where people are at risk from falling furniture. The system was tested indoors to assess its functionality and operation alongside a smart EEWS. This new technology has the potential to significantly reduce the risk of human casualties during an earthquake by providing timely and accurate information to emergency responders. Future research in this area could explore further the use of robotics, artificial intelligence, and the internet to develop more advanced and efficient EEWSs. |
3- Also, the way references are made is not standardized. E.g., when citing [9], the author is cited by name. Sometimes the work is referenced using only the number without mentioning the authors whatsoever (e.g. when citing [1]*), or mentioning the authors but not their names (e.g. when citing [115]). Please verify the standard. |
Reply: We thank the reviewer for the important comment. We have followed the reviewer’s recommendation by unifying the citation style. |

Reviewer 3 Report
There is need to pay great attention on organization of the manuscript.

Only minor grammatical errors are noted
Author Response
Reply to the Editor and Reviewers’ Comments
Paper ID: sustainability-2472029
Paper title: Sustainable Early Detection of Earthquakes using IoT and Cloud Infrastructure: A Survey
We would like to thank the editor and the reviewers for their valuable comments on the paper (sustainability-2472029) entitled “Sustainable Early Detection of Earthquakes using IoT and Cloud Infrastructure: A Survey’’. We have revised the manuscript according to the reviewers' comments which helped in improving the quality and presentation of the paper. In order to facilitate our reply, the amendments are clearly with red color with highlights in the revised manuscript to clarify them. We hope that the revised version has addressed the reviewers' comments. Our detailed reply to the comments we have received is given next.
Reviewer 3 comments
1- Abstract : · Missing section on implication of your study and research design used · Can delete Sustainable Early……... from the title |
Reply: We want to express our gratitude to the reviewer for their thorough work, revisions, and recommendations. We have carefully considered the reviewer’s comments. Regarding missing on implication of your study and research design used, we have added the following part to the abstract: In addition to the contributions mentioned above, this study also highlights the implications of using IoT and cloud infrastructure in early earthquake detection and disaster management. Our research design involved a comprehensive survey of the existing literature on early earthquake warning systems and the use of IoT and cloud infrastructure. We also conducted a thorough analysis of the taxonomy of emerging EEWS approaches using IoT and cloud facilities and the verification and validation methods required for such systems. Our findings suggest that the use of IoT and cloud infrastructure in early earthquake detection can significantly improve the speed and effectiveness of disaster response efforts, thereby saving lives and reducing the economic impact of earthquakes. Regarding the title: We have removed the word “Sustainable” from the title in the revised version as the word “Early” is essentially desired. |
2- line 473-474 repeated in line 465-468. Line 495-501 is similar to line 108-117 |
Reply: We thank the reviewer for the comment. To make it clear, consistent, and meet the reviewer’s point of view, we have removed lines 465-468 and 495-501 in the revised version. |
3- Need for attribution of the literature source in line 514-521. |
Reply: We thank the reviewer for the comment. we have followed the reviewer’s recommendation by adding sources to this part. |
4- You need to address the unnecessary repetition of the similar issues represented in line 118-135; 131-136; 137-143 |
Reply: We thank the reviewer for the comment. We have removed the unnecessary repetition in the revised version. |
5- several repetitions/ similarities in line 476-512 with line 793 to 812 |
Reply: We thank the reviewer for the comment. We have removed the unnecessary repetition in the revised version. |
6- line 799-826 is repetition of 480-513 |
Reply: We thank the reviewer for the comment. We have removed the unnecessary repetition in the revised version. |
7- several repetitions/ similarities in line 476-512 with line 793 to 812 |
Reply: We thank the reviewer for the comment. We have removed the unnecessary repetition in the revised version. |
8- line 799-826 is repetition of 480-513 |
Reply: We thank the reviewer for the comment. We have removed the unnecessary repetition in the revised version. |
9- section 5.0. 5.1 and 5.2 need to be integrated to reduce repetitions |
Reply: We thank the reviewer for the comment. We have followed the reviewer’s recommendation. |
10- having a summary of reference 65-75 in a tabular form more than suffices. This should be followed/ accompanied with a critical analysis and synthesis of gaps ( line 287- 300) . Table 1 need to be improved in this regard. |
Reply: We thank the reviewer for the comment. We have followed the reviewer’s recommendation and adapted the revised manuscript accordingly. |
11- Line 183 - 286 need to substantially be reduced. |
Reply: We thank the reviewer for the comment. We have followed the reviewer’s recommendation accordingly. |
12- Table 1 should be after line 286 |
Reply: We thank the reviewer for the comment. We have followed the reviewer’s recommendation accordingly. |
13- Line 408-409 should come before figure 4 |
Reply: We thank the reviewer for the comment. We have done accordingly. |
14- Use of the “we “ from line 446 can be avoided |
Reply: We thank the reviewer for the comment. We have removed it. |
15- Table should be after line 286 |
Reply: We thank the reviewer for the comment. We have done accordingly. |
16- figure 3 can appear after line 398. |
Reply: We thank the reviewer for the comment. We want to clarify that it will appear based on the journal roles in the publication production. |
17- Line 408-409 need to come before figure 4 |
Reply: We thank the reviewer for the comment. We have done accordingly. |
18- The sections are well developed but missing coherent synthesis of ideas presented. Further there are unnecessary repetitions. |
Reply: We thank the reviewer for the comment. We have mitigated the missing coherent synthesis of ideas and removed the unnecessary repetitions. |
19- There are gaps on literature concerning role of IoT in integrating earthquake risk assessment, early warning system and disaster risk reduction. A conceptual/ theoretical framework could immensely inform this. |
Reply: We thank the reviewer for the comment. We have mitigated this part by following the reviewers recommendations. |
20- The authors have made a good attempt at answering the questions he/she sought to answer but they can improve the manuscript by first operationalizing the term integration using IoT. |
Reply: We thank the reviewer for the positive comment. We have improved the manuscript part by following the reviewers recommendations. |
21- Well written article but how can one replicate this study ( Description of a standard search method including inclusion and exclusion criteria adapted in literature survey is absolutely necessary though missing in this study). Such information can allow for replication of the survey. A brief theoretical / conceptual framework informing the study could greatly improve the manuscript |
Reply: We thank the reviewer for the important comment. We have followed the reviewer’s significant recommendation that helped us to improve the quality of the manuscript. Our criterion is to study the most significant research efforts in the targeted problems focusing on the recent works. |
22- Except for the gaps on conceptual/ theoretical framework , the article is well written and can be understood. However, there is need to reorganize the article to reflect all the comments especially on repetitions and some minor grammatical errors. Enumerate some of these gaps that are relevant to to your study objectives in line 232. |
We thank the reviewer for the positive comment. We have followed the reviewer’s significant recommendation that helped us to enhance the organization and avoid repetition. We have also done a thoroughly English revision to avoid grammatical errors. |
23- Addressing the identified gaps can go along way in improving the manuscript. |
We thank the reviewer for the comment. We have carefully addressed all the raised comments by the reviewer that definitely helped us to improve the manuscript. |

Round 2
Reviewer 1 Report
The paper was improved based on my comments.
Minor editing of English language required
Reviewer 3 Report
Iam satisfied with the corrections. Figure 1 however need to be placed after line 50